# miR-184 modulates *Ilp8* to control developmental timing during normal growth conditions and in response to developmental perturbations

Jervis Fernandes\*, Muhammed Naseem, Ayisha Marwa Mangattu Parambil[‡] and Jishy Varghese[§]

## ABSTRACT

Organismal development depends on the precise coordination of growth and developmental timing, which is regulated by a complex interplay of factors. However, the mechanisms underlying this regulation are not fully understood. Post-transcriptional regulation by microRNAs plays a pivotal role in ensuring the proper timing of gene expression during growth and development. Here, we conducted a genetic screen to identify microRNAs that regulate developmental timing in *Drosophila*. Our screen identified miR-184, previously implicated in germline maturation and embryonic development, as a regulator of pupariation timing by acting in the larval imaginal discs. Using genetic and molecular approaches, we identified *Drosophila Insulin-like peptide 8* (*Dilp8; Ilp8*), a secreted factor that is crucial for regulating developmental stability, as a target of miR-184. During normal larval development, miR-184 facilitates timely pupariation by regulating *Ilp8* levels. Furthermore, we demonstrate that miR-184 plays an essential role in tissue damage responses by aiding in the induction of *Ilp8* expression, which delays pupariation. These findings reveal a previously unreported post-transcriptional regulatory mechanism that links miR-184 to the control of developmental timing under normal growth conditions and in response to tissue damage.

KEY WORDS: Developmental biology, Pupariation timing, Ecdysone signaling, Imaginal discs, *Ilp8*, miR-184, Tissue damage response

## INTRODUCTION

Normal growth and development are essential for the proper functioning and reproductive capacity of an organism. Disruptions in these processes can lead to anomalies such as developmental defects, metabolic imbalances and diseases like cancer. Environmental factors, including nutrition and stress, significantly influence growth and development. In *Drosophila*, growth and developmental timing are tightly linked, with reaching a critical body weight being necessary for the transition from larval to pupal

stages, which triggers adult metamorphosis. Understanding how growth and developmental timing are regulated in *Drosophila* can provide insights into similar mechanisms in other animals, including humans, and suggest potential therapeutic interventions for related developmental disorders.

In *Drosophila*, transitions between larval instars and from larva to pupa are initiated by pulses of the molting hormone Ecdysone (Baehrecke, 1996; Yamanaka et al., 2013). Ecdysone is secreted by the prothoracic gland and converted into its active form, 20-hydroxyecdysone (20-HE), in the larval fat body and other tissues. This hormone initiates a transcriptional cascade via the EcR/USP receptor complex, triggering metamorphosis and halting growth (Yamanaka et al., 2013). The timing of pupariation is governed by Ecdysone secretion; reduced or delayed Ecdysone pulses disrupt development and delay pupariation (Baehrecke, 1996; Yamanaka et al., 2013). Such delays extend the larval feeding period, leading to excessive growth. Environmental factors like temperature, crowding, nutrition, oxygen availability and toxins can modulate pupariation timing (Beamish et al., 2021; Bonnier, 1926; Klepsatel et al., 2018; Layalle et al., 2008; Turingan et al., 2024). Insulin signaling in the prothoracic gland plays a crucial role in Ecdysone production and metamorphosis by modulating the expression of the microRNA bantam (Boulan et al., 2013). Additionally, 20-HE and juvenile hormone (JH) exhibit antagonistic relationships, with JH inhibiting ecdysteroid biosynthesis and being inhibited by 20-HE (Liu et al., 2018). Nutrient stress also modulates Ecdysone secretion via AKH (Adipokinetic hormone, *Drosophila* equivalent of glucagon), which regulates calcium signaling in the prothoracic gland (Hughson et al., 2021). Thus, pupariation timing is controlled by both internal and external factors acting on the prothoracic gland.

A bilateral pair of neurons in the larval central brain release prothoracicotropic hormone (PTTH), which stimulates the prothoracic gland to produce Ecdysone (McBrayer et al., 2007; Ghosh et al., 2010). Loss of the *Ptth* gene leads to delayed pupariation and increased body size due to prolonged feeding. PTTH release is regulated by upstream neuronal inputs that control the timing of PTTH secretion, affecting developmental timing and the onset of metamorphosis (Ghosh et al., 2010; McBrayer et al., 2007). Ecdysteroid biosynthesis involves cytochrome P450 monooxygenases encoded by Halloween genes, which are tightly regulated during Ecdysone synthesis. The transcript levels of Halloween genes correlate well with the Ecdysone titer, implying a stringent transcriptional regulation of these genes during Ecdysone synthesis. Disruptions in Halloween gene expression lead to delayed pupariation (Kamiyama and Niwa, 2022). Additionally, transcriptional regulators that act across various signaling pathways modulate Halloween genes (Danielsen et al., 2014; Kamiyama and Niwa, 2022).

Recent studies also highlight the role of imaginal discs in regulating pupariation timing. Imaginal discs secrete factors like

School of Biology, Indian Institute of Science Education and Research (IISER TVM), Thiruvananthapuram, Kerala, India 695551.
\*Present address: School of Biosciences, The University of Birmingham, Edgbaston, Birmingham B15 2TT, UK. ‡Present address: Institute of Anatomy and Department of Biomedical Research (DBMR), University of Bern, Bern, Switzerland.

§Author for correspondence ( jishy@iisertvm.ac.in)

J.F., 0000-0001-9379-998X; M.N., 0009-0001-8134-6462; A.M.M., 0009-0001-6488-4269; J.V., 0000-0002-5947-3094

ILP8 (*Drosophila* Insulin-like peptide 8), Dpp (Decapentaplegic) and Upd3 (Unpaired 3) to influence pupariation timing. ILP8, a relaxin-like peptide, buffers developmental noise and delays pupariation in response to growth defects in the imaginal discs (Garelli et al., 2012; Colombani et al., 2012). Growth perturbations in the imaginal discs trigger JNK pathway activity, upregulating ILP8 levels in the hemolymph (Colombani et al., 2012; Katsuyama et al., 2015). ILP8 delays pupariation by acting on Lgr3 receptors in the larval brain, which connect PTTH neurons to the prothoracic gland, regulating Ecdysone signaling (Vallejo et al., 2015; Colombani et al., 2015; Garelli et al., 2015; Jaszczak et al., 2016). Furthermore, damage to the larval hindgut increases ILP8 levels, contributing to delayed pupariation (Cohen et al., 2021). Dpp inhibits Ecdysone biosynthesis to prevent premature metamorphosis during imaginal disc growth, while reduced Dpp levels allow progression to metamorphosis once disc growth is complete (Setiawan et al., 2018). Upd3, a cytokine secreted by the imaginal discs, delays pupariation in response to malignant transformations by upregulating bantam and activating JAK/STAT signaling (Romão et al., 2021).

MicroRNAs also play a crucial role in regulating metamorphosis. For example, bantam microRNA in the prothoracic gland modulates pupariation timing by influencing Ecdysone levels (Boulan et al., 2013). Bantam also controls the expression of *jhamt*, a gene involved in JH biosynthesis in the corpus allatum (Qu et al., 2017). miR-8, expressed in the corpus allatum, regulates JH biosynthesis and pupariation timing (Zhang et al., 2021). Numerous microRNAs are differentially modulated by Ecdysone during the larval-pupal transition, highlighting their roles in the regulation of metamorphosis (Jin et al., 2020; Lim et al., 2018). However, the role of microRNAs in regulating the onset of metamorphosis in response to tissue damage has yet to be fully explored.

We report that miR-184, a conserved microRNA that is crucial for oogenesis and early embryonic development in *Drosophila* (Iovino et al., 2009) plays a significant role in regulating developmental timing. *miR-184* mutants exhibit delayed pupariation due to reduced Ecdysone signaling. We show that miR-184 functions in healthy imaginal discs to regulate pupariation timing by targeting *Ilp8* mRNA. Our results also reveal that modulation of miR-184 in response to imaginal disc damage facilitates the delay in larval-to-pupal transition by upregulating *Ilp8* mRNA levels. These findings underscore the essential role of miR-184 in controlling developmental timing both under normal growth conditions and in response to tissue damage.

## RESULTS
### Role of miR-184 in the regulation of pupariation timing
MicroRNAs (miRNAs) are known to regulate various developmental processes, including the timing of developmental transitions. In this study, we explored the role of *Drosophila* miRNAs in regulating the timing of pupariation, a crucial stage when the larva enters the metamorphosis stage under the influence of Ecdysone hormone. We screened 13 microRNA homozygous mutant lines for the mean pupariation time (Table S1). While loss of several miRNAs caused delays in pupariation by 12 h or more, we focused on miR-184, as it is the highest-expressed miRNA in the larvae (FlyAtlas 2.0) and the ring gland (Zhang et al., 2021). We observed that homozygous mutants of *miR-184* exhibited significant delays in puparium formation, indicating that miR-184 is essential for the proper timing of the larval-to-pupal transition (Fig. 1A). To further elucidate the function of miR-184, we employed *miR-184-sponge* (*UAS-miR-184-sp*), a transgenic

inhibitor of miR-184 (Fulga et al., 2015), to reduce miR-184 expression in the entire organism using the *tubGAL4* driver line. Similar to the mutants, downregulation of *miR-184* in the entire organism resulted in a delay in pupariation (Fig. 1B; Fig. S1). The delay in pupariation was rescued by co-overexpression of *miR-184*, confirming the specificity of the *miR-184 sponge* (Fig. S2). These results revealed that miR-184 regulates the timing of pupariation, which prompted us to investigate the regulation of pupariation timing by miR-184.

### miR-184 manages optimal Ecdysone signaling
The timing of pupariation is tightly controlled by the steroid hormone Ecdysone, which is released by the prothoracic gland in developing larvae. Ecdysone is converted into its active form, 20-HE, in peripheral tissues, and regulates the expression of Ecdysone-responsive genes, crucial for pupariation, metamorphosis and adult emergence. Given the pupariation delay observed in miR-184-depleted larvae, we hypothesized that miR-184 might be involved in maintaining optimal Ecdysone signaling.

To test this, we first measured the expression of genes that promote Ecdysone biosynthesis: *Ptth*, *phantom* (*phm*; also known as *phtm*) and *disembodied* (*dib*) in response to downregulation of *miR-184* in the whole larva. We found that miR-184 downregulation led to a significant decrease in the mRNA levels of these genes (Fig. 2A). Furthermore, the transcript levels of key Ecdysone-response genes – *Ecdysone-induced protein 74* (*E74*; *Eip74EF*), *Ecdysone-induced protein 75* (*E75*; *Eip75B*), *Broad-complex* (*BR-C*; *br*) and *fat body protein 1* (*fbp1*) – was also reduced in miR-184-depleted larvae (Fig. 2B). We then proceeded to measure the levels of 20-HE in the late 3rd instar larvae. The level of 20-HE was found to be significantly lower in miR-184-depleted larvae (Fig. 2C). To confirm the role of reduced Ecdysone signaling in the pupariation delay, we fed 20-HE to miR-184-depleted larvae, which completely rescued the pupariation delay (Fig. 2D). These results confirm that miR-184 is required for optimal Ecdysone signaling and the timely transition from larva to pupa. Next, we focused on identifying the tissue where miR-184 acts in regulating Ecdysone signaling.

### miR-184 acts in the imaginal discs to regulate pupariation timing
Although *miR-184* is highly expressed in the ring gland (Zhang et al., 2021), we observed no pupariation delay when miR-184 was downregulated in the prothoracic gland, the primary source of Ecdysone (Fig. 3A). This indicates that the pupariation delay in *miR-184* mutants is not due to miR-184 depletion in the prothoracic gland. Similarly, downregulation of miR-184 in the corpus allatum (CA), which produces JH and negatively regulates Ecdysone, did not affect pupariation timing (Fig. 3B). Moreover, downregulation of miR-184 in the PTTH neurons in the brain, which regulate Ecdysone production, also failed to alter pupariation timing (Fig. 3C). These findings suggest that the role of miR-184 in regulating pupariation timing is not mediated through these tissues. To identify the site of miR-184 activity, we focused on other peripheral tissues where miR-184 could act and regulate Ecdysone signaling.

Recent studies report that imaginal discs in larvae play a role in regulating the timing of pupariation, especially in response to tissue perturbations. When *miR-184* was downregulated in the imaginal discs using the *rnGAL4* driver line, we observed a significant delay in pupariation (Fig. 3D). Furthermore, miR-184 depletion in the imaginal discs reduced the expression of Ecdysone biosynthesis and

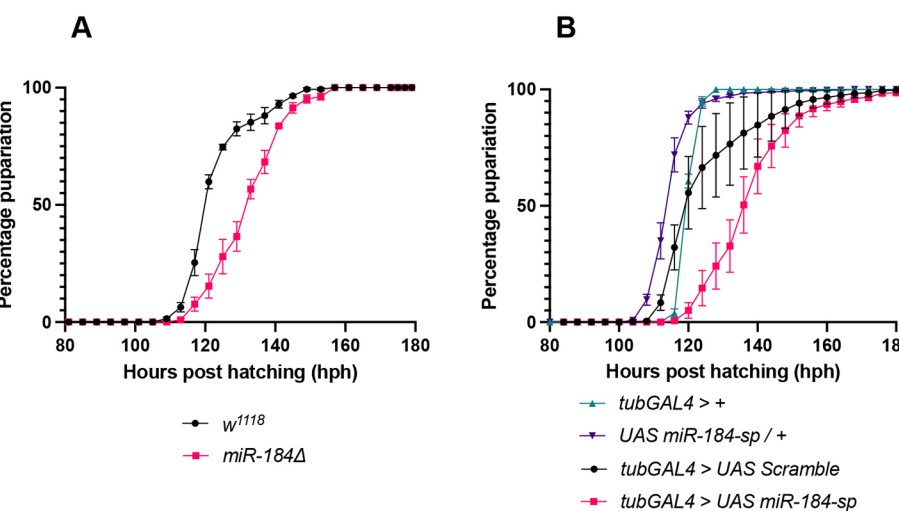

**Fig. 1. miR-184 plays a role in the regulation of pupariation time point.** (A) *miR-184* mutants display delayed pupariation; pupariation rates of *w^1118* (control) (*n*=142) and *miR-184* homozygous null mutants (*n*=104). Data from four independent experiments; *P*<0.0001. (B) Downregulation of miR-184 leads to delayed pupariation; pupariation of *tubGAL4/+* (control) (*n*=155); *UAS-miR-184-sp/+* (control) (*n*=395); *tubGAL4, UAS Scramble* (control) (*n*=955), and *tubGAL4, UAS-miR-184-sp* (*n*=617). Data from five independent experiments [*tubGAL4, UAS Scramble* (control) versus *tubGAL4, UAS-miR-184-sp*]; *P*<0.0001. For A and B, data are mean value of percentage of larvae pupariated, error bars represent s.e.m., *n*=number of larvae analyzed. LogRank test was performed to determine statistical significance.

Ecdysone-response genes (Fig. S3A,B). As *miR-184* expression was reported to be very high in the larval CNS (FlyAtlas 2.0), and to ensure that *rnGAL4* driven miR-184 downregulation phenotype that we saw was not due to expression of GAL4 in the CNS, we checked if the pupariation delay phenotype that we saw was specific to the larval imaginal discs. Expression of *miR-184 sponge* using a pan-neuronal driver *elavGAL4* did not cause any changes to pupariation timing (Fig. S4). These results indicate that miR-184 functions in

the imaginal discs to regulate Ecdysone signaling and the timing of pupariation. As microRNAs function by regulating target gene expression, next we sought to identify miR-184 target genes in the imaginal discs that influence pupariation timing.

### miR-184 modulates pupariation timing via its target *Ilp8*

Our target prediction analysis identified *Ilp8* as a potential target of miR-184 (Fig. S5A). The imaginal discs have been shown to

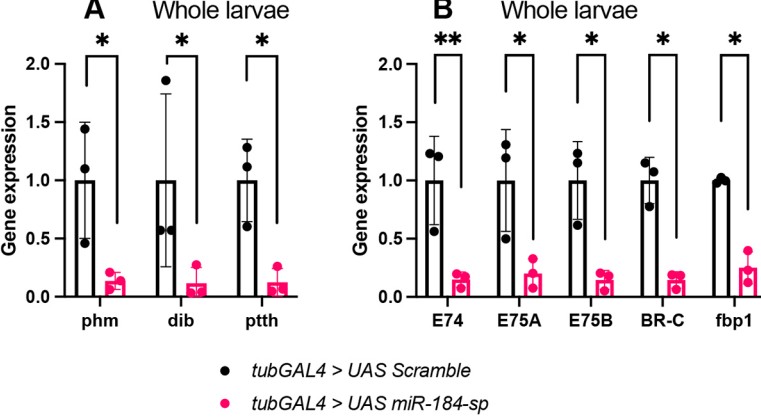

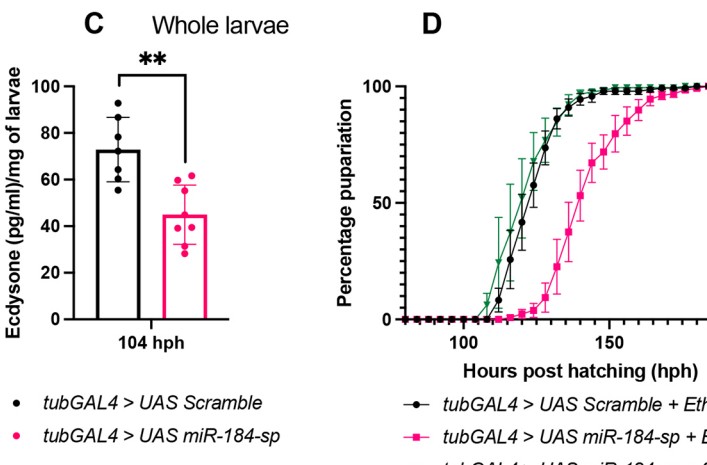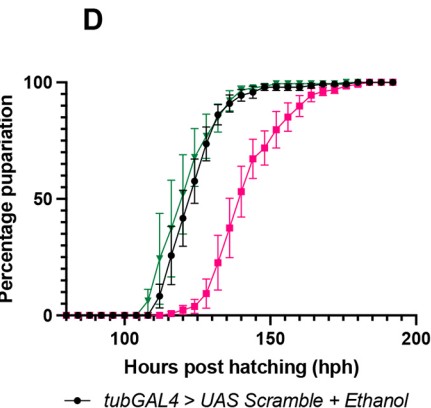

**Fig. 2. miR-184 levels maintain proper Ecdysone signaling.** (A) Expression of genes that promote Ecdysone biosynthesis in response to ubiquitous downregulation of *miR-184*; using *tubGAL4, UAS-miR-184-sp* and *tubGAL4, UAS Scramble* (control). Data from three independent experiments are shown. Fold change of transcript levels of *ptth*, *phm* and *dib* are shown; normalized to *rp49*. The data are normalized to *tubGAL4, UAS Scramble* (control). Total RNA was extracted from whole larvae at 112 h post-hatching. Bar graphs show mean±s.d., with dots denoting absolute values for the replicates; a Welch's *t*-test was performed to determine statistical significance; *P<0.05. (B) Expression of Ecdysone response genes in response to ubiquitous downregulation of miR-184 using *tubGAL4*; *tubGAL4, UAS-scramble* (control) and *tubGAL4, UAS-miR-184-sponge*. Data from three independent experiments are shown. Fold change of transcript levels of *E74*, *E75A*, *E75B*, *BR-C* and *fbp1* are shown; normalized to *rp49*. The data are normalized to *tubGAL4, UAS Scramble* (control). Total RNA was extracted from whole larvae at 112 h post-hatching. Bar graphs show mean±s.d., with dots denoting absolute values for the replicates; a Welch's *t*-test was performed to determine statistical significance; *P<0.05, **P<0.01. (C) Measurement of 20-HE levels (pg/ml)/mg of the larvae at 104 h post-hatching in *tubGAL4, UAS-scramble* (control) and *tubGAL4, UAS-miR-184-sponge*; data from seven independent experiments respectively are shown. Bar graphs show mean±s.d., with dots denoting absolute values for the replicates; a Welch's *t*-test was performed to determine statistical significance; **P<0.01. (D) Pupariation time of *tubGAL4> miR-184-sp* larvae fed 20-HE (0.3 mg/ml) (*P*=0.0748) or ethanol (*P*<0.0001) in comparison to *tubGAL4> control-sp* larvae. Data from five independent experiments are shown, *n*≥128. Data are the mean value of percentage of larvae pupariated, error bars represent s.e.m.. Log rank test was carried out to determine significance.

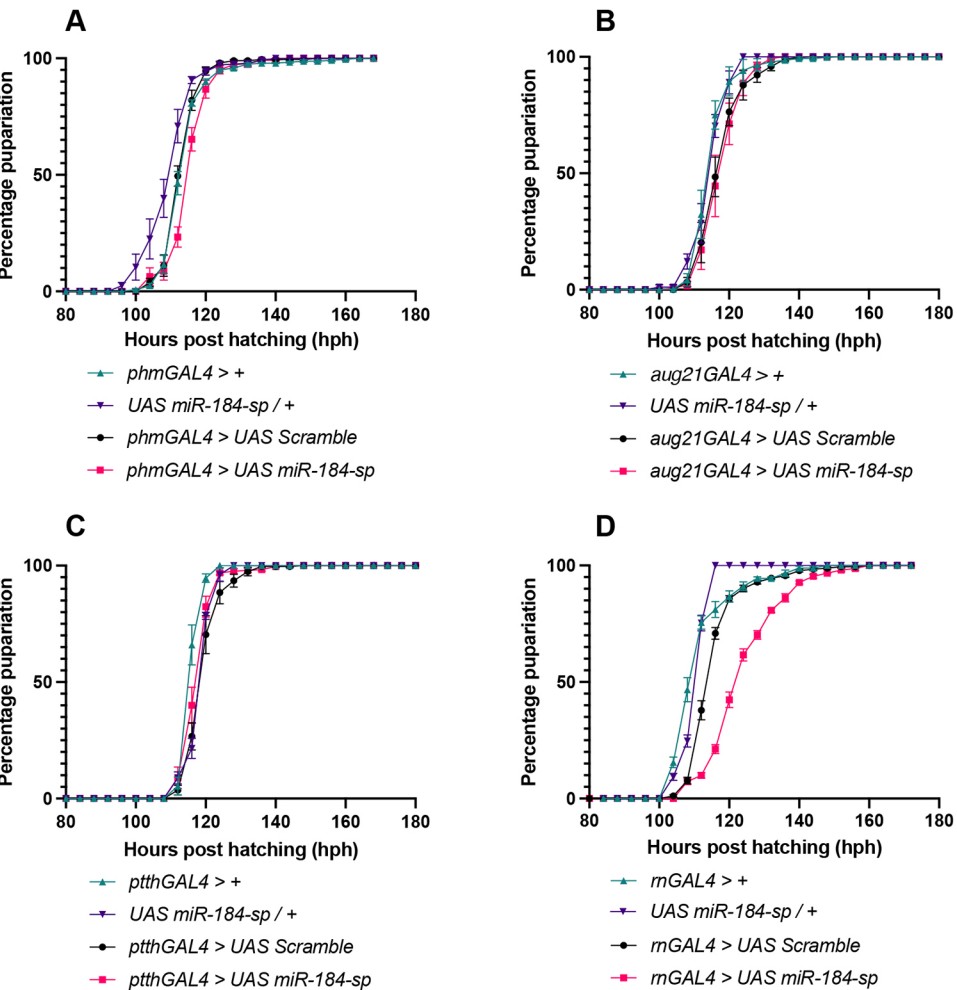

**Fig. 3. miR-184 levels in the imaginal discs regulate pupariation time point.** (A) Downregulation of miR-184 in the prothoracic gland leads to minimal effects on pupariation time point. *phmGAL4, UAS-miR-184-sp*, (*n*=265); *phmGAL4, UAS Scramble* (*n*=206); *phmGAL4/+* (*n*=241); *UAS-miR-184-sp/+* (*n*=248) (controls); five independent experiments are shown. (B) Downregulation of miR-184 in the corpus allatum leads to minimal effects on pupariation time point. *aug21GAL4, UAS-miR-184-sp* (*n*=292) and *aug21GAL4, UAS Scramble* (*n*=601); *aug21GAL4/+* (*n*=408); *UAS-miR-184-sp/+* (*n*=91) (controls); five independent experiments are shown, except for *UAS-miR-184-sp/+* (four sets). (C) Downregulation of miR-184 in the PTTH neurons leads to minimal effects on pupariation time point. *ptthGAL4/+* (*n*=185); *UAS-miR-184-sp/+* (*n*=181); *ptthGAL4, UAS Scramble* (controls) (*n*=277); *ptthGAL4, UAS-miR-184-sp* (*n*=286); five independent experiments are shown, except for *ptthGAL4/+* and *UAS-miR-184-sp/+* (four sets). (D) Downregulation of miR-184 in the imaginal discs leads to a delay in pupariation time point. *rotundGAL4, UAS-miR-184-sponge* (*n*=151); *rotundGAL4, UAS-scramble* (*n*=182); *rnGAL4/+* (*n*=90); *UAS-miR-184-sp/+* (*n*=97) (controls); five independent experiments are shown. *rnGAL4, UAS-miR-184-sponge* versus *rnGAL4, UAS-scramble*; *P*<0.0001. For A-D data are the mean value of percentage of larvae pupariated, error bars represent s.e.m. LogRank test was performed to determine statistical significance.

regulate pupariation timing, especially in response to growth perturbations (Colombani et al., 2012; Garelli et al., 2012; Romão et al., 2021; Setiawan et al., 2018; Simpson et al., 1980; Stieper et al., 2008). One of the key factors involved in this regulation is ILP8, which acts as a developmental checkpoint to delay pupariation in response to growth perturbations (Colombani et al., 2012; Garelli et al., 2012). We hypothesized that miR-184 regulates pupariation timing by modulating *Ilp8* expression.

We found that *Ilp8* mRNA levels were elevated in both *miR-184* knockout larvae (Fig. 4A) and in larvae with whole-body downregulation of miR-184 (Fig. 4B), indicating that *Ilp8* could be a target of miR-184. Furthermore, we observed an increase in the transcript levels of *Ilp8* in response to depleting the expression of miR-184 in the imaginal discs using *rnGAL4* driver (Fig. S5B). This suggests that miR-184 restricts *Ilp8* expression during larval development in the imaginal discs and regulates pupariation timing. To confirm this, we tested whether miR-184 directly regulates *Ilp8* via its 3′ untranslated region (UTR) by using a *tubulin*-promoter-driven *Ilp8-3′UTR-GFP* reporter. We observed reduced GFP expression in the wing imaginal discs in response to *ptcGAL4* driven *miR-184* overexpression, confirming that miR-184 can negatively regulate *Ilp8* via its 3′UTR (Fig. 4C-D′). To confirm the specificity of miR-184 on *Ilp8-3′UTR* we mutated the single miR-184 target site in the *Ilp8*-3′UTR (nucleotides complementary to the miR-184 seed region; Fig. S5A) and created an *Ilp8*-mut-3′ UTR GFP reporter line. The *Ilp8*-mut-3′UTR GFP reporter was not affected by *miR-184* overexpression in the *ptc* expression domain in

the wing disc (Fig. 4E-F′), confirming that the *Ilp8* gene is a target for post-transcriptional regulation by miR-184.

We then tested whether elevated *Ilp8* levels in *miR-184* mutants were responsible for the pupariation delay. Indeed, reducing *Ilp8* levels in *miR-184* mutant larvae by replacing one copy of wild-type *Ilp8* with the hypomorphic *Ilp8* mutant allele (*Ilp8^{MI00727}*) rescued the pupariation delay (Fig. 4G). Similarly, co-expression of *Ilp8* RNAi with *miR-184 sponge* also rescued the pupariation timing (Fig. 4H). These results confirm that miR-184 regulates pupariation timing by controlling *Ilp8* expression during normal development.

## miR-184 regulates a developmental checkpoint in response to perturbations

In response to developmental stresses, such as excessive cell death or hyperplastic growth, *Ilp8* expression is upregulated, which suppresses Ecdysone synthesis, delaying pupariation to allow for tissue repair and growth coordination (Garelli et al., 2012; Colombani et al., 2012). To investigate whether miR-184 contributes to developmental stress responses by aiding in ILP8 induction, we induced tissue damage in the imaginal discs using a cold-sensitive version of the type 2 ribosome-inactivating protein Ricin-A (RA^{CS}), which causes cell death and elevates *Ilp8* expression (Sanchez et al., 2019). *RA^{CS}* expression in the imaginal discs led to increased *Ilp8* transcript levels and a significant delay in pupariation (Fig. S6A-C). We also observed that *RA^{CS}* expression activated JNK signaling, a known pathway involved in the stress response and elevation of *Ilp8* expression (Colombani et al., 2012; Sanchez et al., 2019), by assessing the

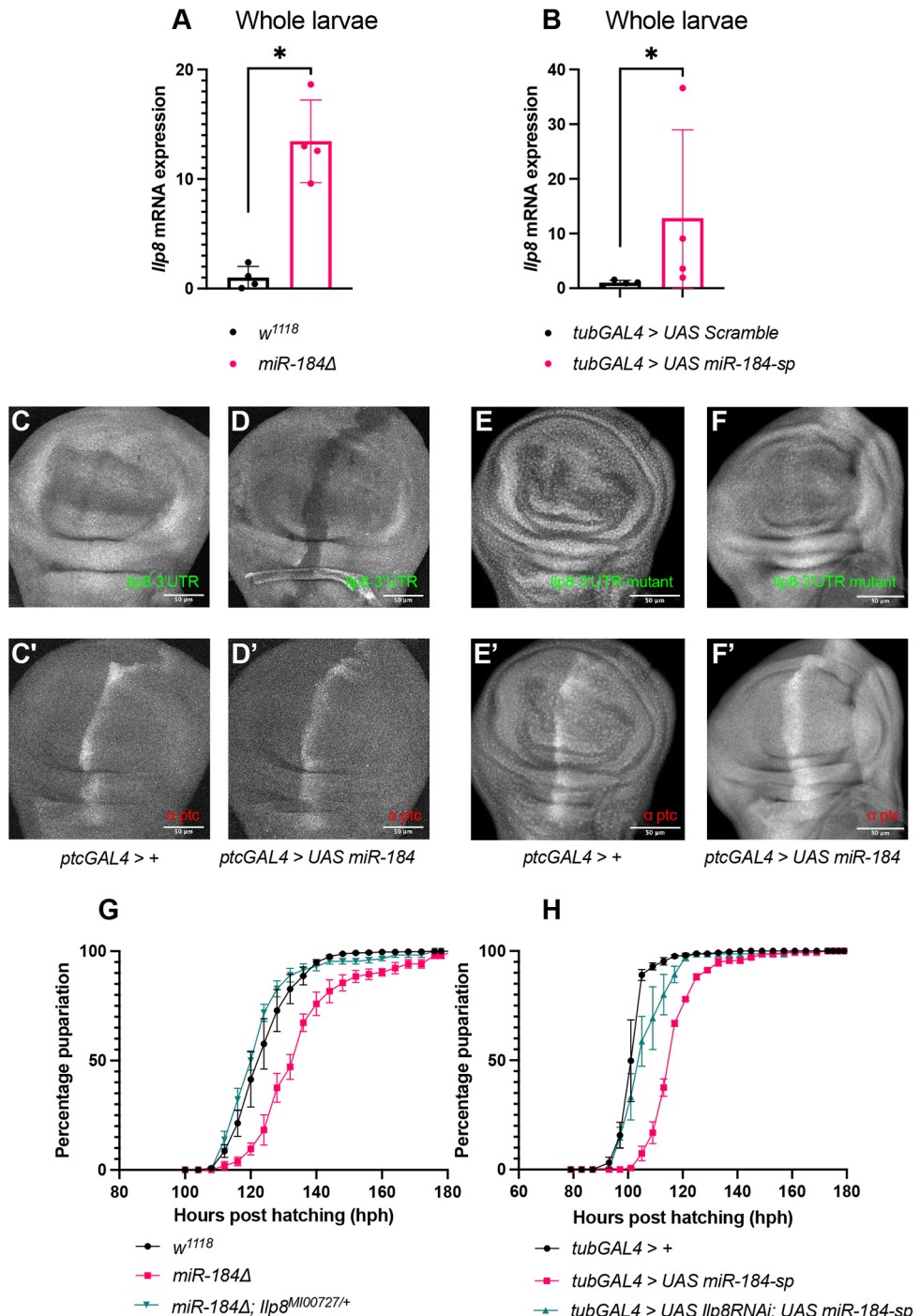

**Fig. 4. miR-184 regulates developmental time point via its target gene *Ilp8*.** (A) *Ilp8* mRNA levels in *w1118* (control) and *miR-184* homozygous null mutants from four independent experiments are shown; *$P$=0.0281. Fold change of transcript levels of *Ilp8* is shown, normalized to *rp49*. The data are normalized to *w1118* (control). Total RNA was extracted from whole larvae at 112 h post-hatching. (B) *Ilp8* mRNA levels in *tubGAL4, UAS Scramble* (control) and *tubGAL4, UAS-miR-184-sp*; four independent experiments are shown; *$P$=0.0428. Fold change of transcript levels of *Ilp8* is shown, normalized to *rp49*. The data are normalized to *tubGAL4, UAS Scramble* (control). Total RNA was extracted from whole larvae at 96 h post-hatching. For A and B, bar graphs show mean±s.d., with dots denoting absolute values for the replicates. Welch's *t*-test was performed to determine statistical significance. (C,C′) *Ilp8*-3′UTR-GFP reporter assay; reporter GFP expression (anti-GFP, labeled in green; C) and patched expression (anti-Ptc, labelled in red; C′) in *ptcGAL4* (control) larval wing discs. (D,D′) *Ilp8*-3′UTR-GFP reporter assay; reporter GFP expression (anti-GFP; D) and patched expression (anti-Ptc; D′) in *ptcGAL4, UAS-miR-184* (overexpression) wing discs. (E,E′) *Ilp8*-3′UTR mutant-GFP reporter assay; mut-3′UTR reporter GFP expression (anti-GFP; E) and patched expression (anti-Ptc; E′) in *ptcGAL4* (control) wing disc. (F,F′) *Ilp8*-3′UTR mutant-GFP reporter assay; mut-3′UTR reporter GFP expression (anti-GFP; F) and patched expression (anti-Ptc; F′) in *ptcGAL4, UAS-miR-184* (overexpression). (G) Rescue of pupariation delay; *w1118* (control) (*n*=486), *miR-184−/−* (homozygous mutant) (*n*=104) (*P*<0.0001) and *miR-184−/−; Ilp8^MI00727/+* (rescue) (*n*=173); *P*=0.0438; five independent experiments are shown. (H) Rescue of pupariation delay; *tubGAL4, UAS Scramble* (control) (*n*=255) and *tubGAL4, UAS-miR-184-sp* (miR-184 downregulation) (*n*=136) and *tubGAL4, UAS-miR-184-sp, Ilp8-RNAi* (rescue) (*n*=76); *P*<0.0001; five independent experiments are shown. For G and H, data are mean value of percentage of larvae pupariated, error bars represent s.e.m. *P*-values are calculated in comparison to the controls. LogRank test was performed to determine statistical significance.

expression of *puc* (*puckered*) and *Mmp1* (*matrix metalloproteinase 1*) – two target genes of JNK signaling – in the wing imaginal discs (Fig. S6D,E).

Despite the well-documented dynamic expression pattern of *miR-184* during larval development and its regulation by nutrient availability and tumorous conditions (Fernandes et al., 2022; Shu et al., 2017), its modulation during developmental perturbations remains unexplored. Our findings imply that miR-184 plays a role in inducing *Ilp8* expression in response to such perturbations. We tested this by first measuring the levels of miR-184 in response to tissue damage induced by *RA^{CS}* expression. Remarkably, *RA^{CS}* expression led to a decrease in miR-184 levels, together with increased *Ilp8* expression in the wing imaginal discs (Fig. 5A; Fig. S6B), suggesting that miR-184 plays a role in the responses to developmental stress. However, in comparison to the many-fold induction of *Ilp8* mRNA levels in response to tissue damage (Fig. S6B), the reduction of miR-184 is modest, suggesting the role of transcriptional upregulation of *Ilp8*. To check if the regulation of *Ilp8* in response to tissue damage by *RA^{CS}* expression is at a post-transcriptional level, we tested the *dip8-3′UTR* reporter GFP in this background. A moderate increase of the GFP reporter was found in response to *ptcGAL4*-mediated overexpression of *RA^{CS}* (Fig. S6F-G′). Increase of the *Ilp8*-3′UTR GFP reporter in response

to tissue damage further suggests post-transcriptional regulation by miR-184 contributing to the induction of *Ilp8* expression together with possible other transcriptional means. Nevertheless, we cannot rule out the possibility that the miR-184-RNA-induced silencing complex (RISC) is reused multiple times to regulate multiple *Ilp8* messengers, and the reduction of miR-184 accounts for the majority of the *Ilp8* mRNA induction.

To confirm that reduced miR-184 expression contributes to *Ilp8* upregulation and the pupariation delay under stress, we manipulated miR-184 expression in *RA^{CS}*-induced conditions. As a control, we downregulated *Ilp8* using RNAi in the *RA^{CS}* overexpression background, which reduced *Ilp8* expression (Fig. 5C) and rescued the pupariation delay (Fig. 5B). Co-overexpression of *miR-184* with *RA^{CS}* suppressed *Ilp8* expression (Fig. 5C) and rescued the pupariation delay (Fig. 5B), confirming that reduced miR-184 expression contributes to enhanced *Ilp8* levels and pupariation delay under tissue stress. Thus, our results show that developmental tissue damage responses act in part through the regulation of miR-184.

Finally, to investigate the possible signaling mechanisms that regulate miR-184 expression under stress, we inhibited endogenous JNK signaling in the larval imaginal discs using the dominant-negative form of *basket*, *Drosophila* JNK (*bskDN*), based on previous reports that the JNK pathway is involved in tissue damage

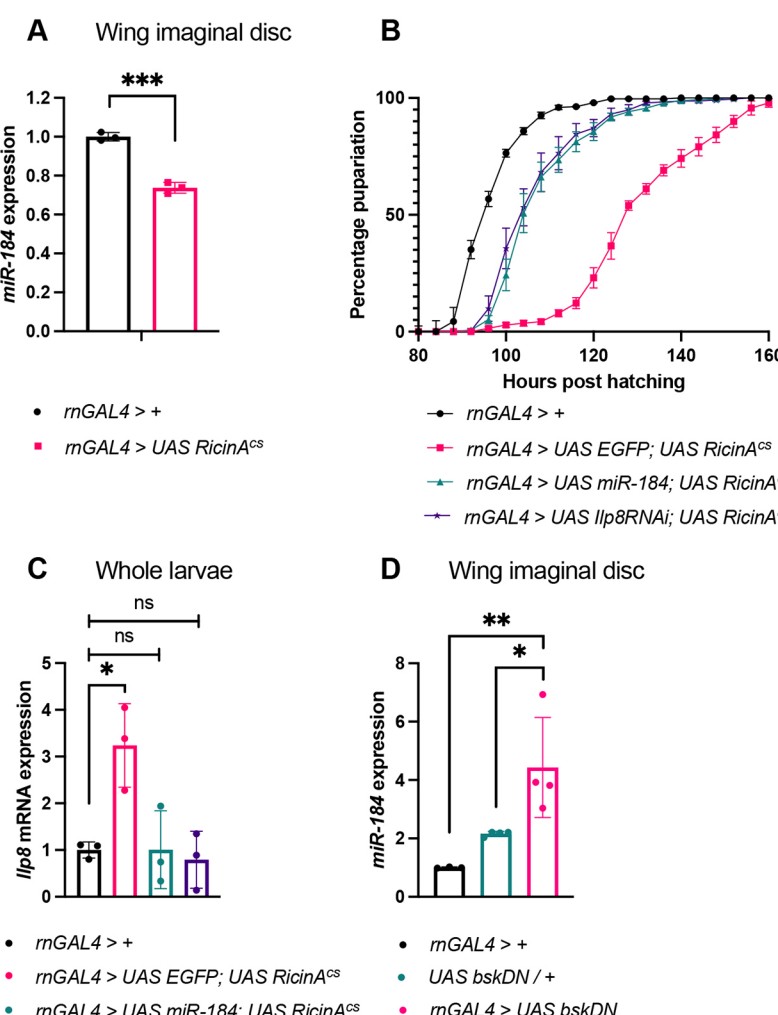

**Fig. 5. Role of miR-184 in Ricin-induced developmental perturbation.** (A) miR-184 levels are affected in response to Ricin-A expression in the imaginal discs; *rnGAL4/+* (control) and *rnGAL4, UAS-RA^{cs}*; three independent experiments are shown; *P*=0.001. Fold change of miR-184 is shown, normalized to *2SrRNA*.Total RNA was extracted from the wing imaginal discs at 100 h post-hatching. Bar graph shows mean±s.d., with dots denoting absolute values for the replicates; a Welch's *t*-test was performed to determine statistical significance; ****P*<0.001. The data are normalized to *rnGAL4/+* (control). (B) Rescue of pupariation delays in response to Ricin-A expression in the imaginal discs, *rnGAL4* (control) (*n*=298); *rnGAL4, UAS-EGFP, UAS-RA^{cs}* (*n*=139) (*P*<0.0001); *rnGAL4, UAS-miR-184-OE, UAS-RA^{cs}* (*n*=272) (*P*<0.0001); *rnGAL4, UAS-Ilp8-RNAi, UAS-RA^{cs}* (*n*=233) (*P*<0.0001). Five independent experiments are shown. Data are mean value of percentage of larvae pupariated, error bars represent s.e.m., *n*=number of larvae analyzed. LogRank test was performed to determine statistical significance, *P*-values are calculated in comparison to *rnGAL4* (control). (C) Rescue of *Ilp8* levels in the wing imaginal discs expressing Ricin-A: *rnGAL4* (control); *rnGAL4, UAS-EGFP, UAS-RA^{cs}* (**P*=0.0383); *rnGAL4, UAS-Ilp8-RNAi, UAS-RA^{cs}* (*P*=0.9224; ns, not significant); *rnGAL4, UAS-miR-184-OE, UAS-RA^{cs}* (*P*=0.9945). Three independent experiments are shown. Fold change of transcript levels of *Ilp8* is shown, normalized to *rp49*. The data are normalized to *rnGAL4/+* (control). Total RNA was extracted from whole larvae at 100 h post-hatching. Bar graph shows mean±s.d., with dots denoting absolute values for the replicates. A Brown-Forsythe and Welch's ANOVA test was performed to determine statistical significance, *P*-values in comparison to *rnGAL4* (control). (D) Blocking JNK signaling in the imaginal discs leads to the induction of miR-184 levels in *rnGAL4, UAS-bskDN*; against controls *rnGAL4/+* (***P*=0.0037); and *UAS-bskDN/+* (**P*=0.0308); four independent experiments are shown (three replicates for *rnGAL4/+*). Fold change of miR-184 is shown, normalized to *2SrRNA*. The data are normalized to *rnGAL4/+* (control). Total RNA was extracted from the wing imaginal disc at 100 h post-hatching. Bar graph shows mean±s.d., with dots denoting absolute values for the replicates. Unpaired two-tailed Student's *t*-test was performed to determine statistical significance.

responses in regulating growth and development (Colombani et al., 2012; Sanchez et al., 2019). We found that blocking JNK signaling increased miR-184 levels in the wing imaginal discs (Fig. 5D). This confirms the role of JNK signaling in the regulation of miR-184 levels and suggests that induction of JNK signaling during tissue damage leads to reduction of miR-184 levels (Fig. 5A), which contributes to induction of *Ilp8* levels and pupariation delays.

## DISCUSSION

Achieving the final body size of an organism, as determined by its genetic makeup, requires precise coordination between growth and developmental timing. Environmental factors, which influence gene expression and signaling pathways that regulate these processes, introduce additional variability. To buffer against environmental fluctuations, biological systems employ robust mechanisms, including the post-transcriptional regulation of gene expression by microRNAs. A genetic screen conducted in our lab identified the microRNA, miR-184 as a regulator of pupariation timing. Among several *Drosophila* microRNAs, miR-184 is a maternally deposited microRNA (Iovino et al., 2009) with a highly dynamic expression pattern across various developmental stages (Li et al., 2011).

Here, we show that miR-184 is essential for normal Ecdysone signaling and regulates the developmental transition from larval to pupal stages. The delayed pupariation observed in *miR-184* mutants was associated with reduced Ecdysone signaling. Further, we show that miR-184 acts in the imaginal discs to control Ecdysone levels and pupariation timing. Our experiments identified *Ilp8*, a gene encoding a secreted factor that stabilizes organ and body size (Colombani et al., 2012; Garelli et al., 2012), as a direct target of miR-184. ILP8 delays pupariation by modulating Ecdysone signaling during imaginal disc damage (Colombani et al., 2012; Garelli et al., 2012). Loss of *miR-184* resulted in elevated *Ilp8* transcript levels, while suppressing *Ilp8* in this context rescued the pupariation delay. Furthermore, we show that miR-184 directly regulates *Ilp8* expression via its 3′UTR post-transcriptionally. Our findings confirm that *Ilp8* levels are tightly controlled by miR-184 during normal larval development, with the loss of this regulation causing delays in pupariation.

Previous studies have highlighted the importance of maintaining optimal *Ilp8* levels during development, as elevated *Ilp8* expression is associated with pupariation delays. The role of ILP8 in ensuring developmental stability was first reported by Garelli et al. (2012) and has been corroborated later by several groups (Colombani et al., 2015; Garelli et al., 2015; Boone et al., 2016; Blanco-Obregon et al., 2022). Notably, loss of *Ilp8* increased wing asymmetry, underscoring its role in buffering developmental noise (Garelli et al., 2012, 2015; Colombani et al., 2015; Boone et al., 2016; Blanco-Obregon et al., 2022). Thus, we report a previously undescribed post-transcriptional mechanism by which miR-184 regulates *Ilp8* expression, a process essential for maintaining proper developmental timing.

The Hippo signaling pathway plays a pivotal role in maintaining *Ilp8* expression during normal development, ensuring symmetric tissue growth (Boone et al., 2016). Additionally, during pupariation, Ecdysone-triggered *Ilp8* signaling from the cuticle epidermis orchestrates behavioral and morphogenetic transitions (Heredia et al., 2021). The post-transcriptional regulation of *Ilp8* by Pacman/ XRN1 (Jones et al., 2016) are also essential for maintaining appropriate *Ilp8* levels during development. In addition, it was reported that, during ribosome stress in the larval imaginal discs, the stress-response transcription factor Xrp1 acts through *Ilp8* in regulating systemic growth (Boulan et al., 2019; Destefanis et al., 2022). Building on this foundational work, our study identifies a

post-transcriptional regulatory mechanism where miR-184 precisely modulates *Ilp8* expression, ensuring proper timing of crucial developmental transitions.

Lgr3, the receptor for ILP8, is expressed in a small subset of neurons in the larval brain and plays a pivotal role in mediating developmental delays caused by ILP8 in response to tissue damage (Vallejo et al., 2015; Colombani et al., 2015; Garelli et al., 2015). The Lgr3 neurons directly interact with PTTH-producing neurons and insulin-producing neurons, forming a regulatory circuit that controls developmental timing and growth. Through the Lgr3-PTTH neuronal circuitry, ILP8 modulates Ecdysone synthesis in the prothoracic gland, inducing developmental delays in response to tissue damage (Vallejo et al., 2015; Colombani et al., 2015; Jaszczak et al., 2016). In this study, we show that the loss of *miR-184* elevates *Ilp8* levels, which in turn reduces the expression of PTTH and disrupts Ecdysone signaling. This cascade leads to the pupariation delays observed in *miR-184* mutants.

Recent studies have shown that, in response to growth perturbations, ILP8 activates a developmental checkpoint that delays developmental timing and slows the growth of unaffected tissues to maintain overall organismal stability. Several signaling pathways regulate *Ilp8* expression during tissue damage or stress. In early regenerating discs, JAK/STAT signaling induces *Ilp8* expression, delaying the onset of pupariation in response to disc fragmentation (Katsuyama et al., 2015). Similarly, malignant disc tumors exhibit enhanced secretion of ILP8, regulated by JNK signaling (Romão et al., 2021). Blocking JNK signaling in damaged imaginal discs disrupts *Ilp8* induction and alleviates eclosion delays or regulates cell growth (Colombani et al., 2012; Sanchez et al., 2019). In this study, we found that tissue damage leads to a reduction in miR-184 levels, likely mediated by JNK activation, which in turn results in a post-transcriptional increase in *Ilp8* expression and contributes to developmental delays. However, our experiments also suggest molecular mechanisms other than post-transcriptional regulation by miR-184 in the induction of *Ilp8* that results in pupariation delays, which needs further investigation. Additionally, it remains to be explored if expression of miR-184 in response to tissue damage is regulated by other signaling mechanisms, as mentioned above (Katsuyama et al., 2015; Jones et al., 2016; Heredia et al., 2021; Boulan et al., 2019; Destefanis et al., 2022), leading to enhanced *Ilp8* levels and pupariation delays.

miR-184 plays a crucial role in the female germline and early developmental processes (Iovino et al., 2009). miR-184 has also been identified as a nutrient-sensitive microRNA, with increased expression under low-nutrient conditions, in which it is essential for survival (Fernandes et al., 2022; Gendron and Pletcher, 2017). Notably, overexpression of the epithelial tricellular junction protein Gliotactin upregulates miR-184 levels via BMP signaling, establishing a feedback mechanism for its regulation (Sharifkhodaei et al., 2016). Furthermore, miR-184 levels were observed to decrease in wing disc tumors (Shu et al., 2017), though the functional implications of this reduction remained unclear. These findings underscore the diverse regulatory roles of miR-184 in both normal and stress-induced developmental contexts.

In addition to reports that *Ilp8* levels in the imaginal discs of *Drosophila* are crucial to tissue damage responses and serve as a critical developmental checkpoint (Colombani et al., 2012; Garelli et al., 2012), several studies have attributed additional functions for *Ilp8*. During regenerative growth, Ilp8, dMyc and Wg levels are constrained by the *brain tumor* (*brat*) gene in regenerating tissues to ensure proper patterning, cell-fate specification and differentiation (Abidi et al., 2023). In early disc regeneration, *Ilp8* expression is

induced, which controls developmental timing to promote damage recovery (Katsuyama et al., 2015). Additionally, ILP8 plays a pivotal role during metamorphosis by maintaining the symmetry of adult organs (Boone et al., 2016; Garelli et al., 2012). Recent studies have demonstrated that ILP8, through its receptor Drl, is involved in the transdetermination of imaginal discs, highlighting its broader developmental significance (Nemoto et al., 2023). In adult *Drosophila*, *Ilp8* levels in the ovarian follicle cells regulate ovulation (Liao and Nässel, 2020) and influence responses to starvation and desiccation. While our study establishes the role of miR-184 in regulating *Ilp8* in the context of imaginal disc damage, it remains an open question whether this regulatory axis extends to other developmental or physiological contexts. These findings suggest that miR-184-mediated control of *Ilp8* may have broader implications in growth, regeneration and tissue homeostasis across diverse systems.

Our study proposes a mechanism whereby miR-184 fine-tunes *Ilp8* levels to coordinate developmental timing by maintaining appropriate Ecdysone signaling under normal conditions and actively adjusting timing during perturbations to facilitate damage repair (Fig. 6A,B). These findings highlight the importance of post-transcriptional regulation in responding to developmental disturbances, opening avenues for future research on tissue-specific and cross-species mechanisms of microRNA-mediated regulation.

## MATERIALS AND METHODS
### Fly stocks
The fly stocks and experimental crosses were reared in food containing standard laboratory cornmeal medium in an incubator maintained at 25°C, 70% humidity, and a 12 h light: 12 h dark cycle. As recommended for cold sensitive *RA*, all experiments with *RA$^{cs}$* were performed at 29°C (permissive temperature) (Sanchez et al., 2019). One liter of the *Drosophila* medium contains 58.2 g cornmeal, 50.8 g dextrose, 23.6 g yeast, 8 g agar, and 3 g nipagin (10% in 100% ethanol). *w$^{1118}$* (6326) and *UAS-miR-184-sponge*

(61396) lines were obtained from Bloomington *Drosophila* Stock Center (BDSC). The *miR-184$^{KO}$* line (116326) was obtained from *Drosophila* Genomics Resource Center (DGRC). The *UAS-Scramble-sponge* line was a kind gift from Dr Tudor Fulga (University of Oxford, England). The *UAS-Ilp8 RNAi* and *Ilp8$^{MI00727}$* lines were a kind gift from Dr Pierre Leopold's lab (Institut Curie, France). The *tubGAL4, phmGAL4, aug21GAL4, miR-10$^{KO}$*, *miR-124$^{KO}$, miR-133$^{KO}$, miR-285$^{KO}$, miR-283$^{KO}$, miR-304$^{KO}$, miR-210$^{KO}$*, *miR-219$^{KO}$, miR-31b$^{KO}$, miR-137$^{KO}$, miR-375$^{KO}$* and *miR-193$^{KO}$* lines were a kind gift from Dr Stephen Cohen's lab (University of Copenhagen, Denmark). The *rnGAL4* and *UAS-RA$^{cs}$* lines were a kind gift from Dr Marco Milan's lab (Institute for Research in Biomedicine, Barcelona). The *ptthGAL4* line was a kind gift from Dr Nisha Kannan's lab (IISER TVM, India). The *UAS-miR-184* line was a kind gift from Dr Kweon Yu's lab (Korea Research Institute of Bioscience and Biotechnology, Daejeon, South Korea). The *tub-GFP-Ilp8-3′UTR* reporter line was created by cloning the entire 231 bp *Ilp8-3′UTR* downstream of EGFP under the control of *tubulin* promoter using a modified pCasper4 plasmid (Varghese and Cohen, 2007). The *Ilp8-3′UTR* forward primer 5′-AAAAAATCTAGAGAGAGCT-CCAGTGTCATGC-3′ and *Ilp8-3′UTR* reverse primer 5′-AAAAAACTC-GAGCCGATCAGTTGCCGTATTCG-3′ were used to clone the *Ilp8-3′UTR* using the restriction enzymes *Not*1 (1166A) and *Xho*1 (1094A) from Takara Bio Inc. The plasmid was microinjected into *w$^{1118}$* embryos by the National Centre for Biological Sciences, Tata Institute of Fundamental Research, Bengaluru *Drosophila* facility for generating random P-element insertion lines and their progeny were screened to obtain the *tub-GFP-Ilp8-3′UTR* balanced line. The *Ilp8-mut-3′UTR* was generated by introducing a change of miR-184 target site (sequence complementary to the 8-mer seed region of miR-184 in the 3′UTR sequence) from UCCGUCCA to AGGCAGGU. The 231 bp *Ilp8-mut-3′UTR* was synthesized by SLV Scientific and cloned downstream of EGFP under the control of *tubulin* promoter using a modified pCasper4 plasmid to generate the *tub-GFP-Ilp8-mut-3′UTR*. All plasmids and transgenic lines generated for this study are available upon email request to the corresponding author.

### Pupariation time point assay
Pupariation time point assay measures the time taken from hatching to pupariation. All the larvae hatched before the peak hatching are picked and

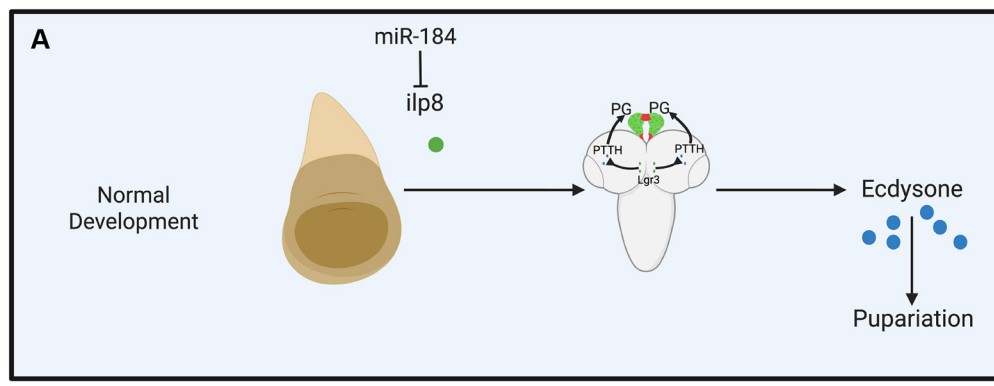

**Fig. 6. Model showing the function of miR-184 during growth and development.** (A,B) miR-184 regulates *Ilp8* expression to synchronize developmental timing by ensuring optimal Ecdysone signaling under normal conditions (A) and by dynamically modifying this timing in response to perturbations to promote damage repair (B). Image was created in BioRender by Varghese, J., 2026. https://BioRender.com/l1npzt2. This figure was sublicensed under CC-BY 4.0 terms.

discarded. The remaining eggs are kept in a 25°C incubator for 1 h. Fifty first instar larvae were collected within a tight time window of 1 h and reared in a 25°C incubator for the next few days, with the number of larvae pupariated being quantified every 4 h. LogRank statistical analyses were carried out in Oasis2 (Yang et al., 2011). Pupariation experiments involving the use of *UAS-RA^{cs}* were carried out at 29°C.

### Ecdysone feeding experiment
L1 larvae were reared in normal media at 25°C till 80 h post-hatching. At 80 h post-hatching, 25 larvae were transferred to either 20-HE (0.3 mg/ml) (Sigma-Aldrich, H5142) or ethanol media and maintained at 25°C. The number of pupae was recorded every 4 h.

### Ecdysone quantification
For Ecdysone extraction, 15 larvae were collected in 2 ml safe-lock Eppendorf tubes, flash frozen in liquid nitrogen, and stored at −80°C at the indicated post-hatching time point. The samples were homogenized using 0.1 mm zirconium oxide beads and the Bullet Blender Storm (BBY24M, Next Advance) in 300 µl methanol (Sigma-Aldrich, 322415), centrifuged at 15,000 rpm (21,130 *g*) for 5 min at 4°C, following which the supernatant was transferred to a new microcentrifuge tube. Then 300 µl of methanol was added to the samples and vortex mixed before centrifuging in an Eppendorf centrifuge 5424 R and transferring the supernatant to the same tube. Lastly, the procedure was repeated with 300 µl of ethanol (Himedia, MB228) to obtain a final volume of 900 µl of the supernatant. The samples containing the supernatant were centrifuged at 15,000 rpm (21,130 *g*) for 5 min at 4°C in multiple rounds while transferring the supernatant to remove any remaining debris. The samples were evaporated using an Eppendorf Speedvac centrifuge and the pellet was dissolved in 200 µl of ELISA buffer (EIA Buffer). The samples were stored at room temperature for 2 h with intermediate vigorous vortexing to ensure proper dissolution of the pellet. ELISA was performed as per the instructions of the 20-Hydroxyecdysone ELISA kit (Cayman Chemicals, 501390). The readings were taken at 405 nm using the TECAN Infinite M200 pro-multimode plate reader. The data was analyzed as per the manufacturer's instructions.

### Sample collection for RNA isolation
Larval samples were collected 112 h post-hatching, and at 100 h for the RA^{cs} experiments, and plunged into liquid nitrogen. The samples were then transferred to −80°C for storage. For the imaginal discs RNA isolation: 20 larvae per set were collected at 92∼100 h post-hatching and dissected in PBS solution. The wing imaginal discs were isolated and stored in TriZol immediately post-dissection before proceeding for total RNA extraction.

### qRT-PCR
Total RNA was extracted from samples each using the TriZol Reagent, as per the manufacturer's instructions, and pelleted using chloroform and isopropanol. The pellet was washed with 70% ethanol and centrifuged at 15,000 rpm (21,130 *g*). The pellet was dried and dissolved in 40 µl of molecular biology-grade water (Himedia). Any contaminating DNA was degraded with the help of a Qiagen RNase-Free DNase set as per the manufacturer's instructions. Following this, the RNA was re-extracted and dissolved in 40 µl of molecular biology-grade water (Himedia). Takara Perfect Real-time kit was used to convert similar quantities of the RNA to cDNA. The cDNA was used for qPCR quantification using TB Green Premix E×Taq (Tli RNaseH Plus) in a CFX96 machine (Bio-Rad). The level of the genes was normalized to the level of *rp49* (*RpL32*). The ΔCt values generated were analyzed using the unpaired two-tailed Student's *t*-test in GraphPad Prism. The mRNA fold change was plotted by using $2^{-\Delta\Delta Ct}$ values.

### microRNA qRT-PCR
Total RNA was extracted from L3 larvae using the TriZol extraction method (described above). Total RNA was polyadenylated and reverse transcribed using poly(A) polymerase and SMART MMLV reverse transcriptase supplied with the Mir-X miRNA qRT-PCR TB Green Kit. The TB Green Advantage qPCR Premix and mRQ 3′ primer were then used in real-time

qPCR, along with miR-184-specific 5′ primer(s), to quantify specific miRNA sequences in the cDNA. The levels of the microRNA were normalized to the levels of *2SrRNA*. The ΔCt values generated were analyzed by the unpaired two-tailed Student's *t*-test. The miRNA fold change was plotted by using $2^{-\Delta\Delta Ct}$ values. Primers used for qRT-PCR are shown in Table S2.

### Antibody stainings and immunofluorescence
For every genotype, non-wandering early third instar larvae were dissected in 1× ice-cold PBS and fixed in 4% paraformaldehyde (PFA) (Sigma-Aldrich, P6148) solution for 20 min at room temperature. Following the removal of PFA, the tissues were washed using PBT [1× phosphate-buffered saline+0.1% Triton X-100 (Bio-Rad, 161-0407)]. After adding the blocking solution [PBT+0.1% bovine serum albumin (Sigma-Aldrich, A2153)], the samples were allowed to sit at room temperature for 45 min on a nutator to ensure efficient blocking. The samples were then incubated with anti-ptc antibody (Developmental Studies Hybridoma Bank, 1:200 dilution with blocking solution) and anti-GFP antibody (Invitrogen, A-6455, polyclonal antibody, 1:500 dilution with blocking solution) overnight at 4°C while being rotated continuously. Following a thorough PBT wash, the tissues were left to incubate for 2 h at room temperature with the secondary antibody: anti-mouse IgG-conjugated with Alex-Fluor 633 (Invitrogen, A-21052) and anti-rabbit IgG-conjugated with Alex-Fluor 488 (Invitrogen, A-11008) diluted 1:500 with blocking solution were used to fluorescently label the samples, respectively. The samples were thoroughly cleaned and mounted using a drop of SlowFade Gold Antifade Reagent (Thermo Fisher Scientific, S36939) after 2 h. The samples were visualized and imaged by a confocal microscope (Leica DM6000B). ImageJ software was used to analyze the images further.

### Bioinformatics
microRNA targets were predicted using the target prediction software TargetScan Fly 7.2 and DIANA-microT-CDS. STarMir was further used to confirm and generate the RNA hybrid (Paraskevopoulou et al., 2013; Rennie et al., 2014).

#### Acknowledgements
We thank Dr Smitha Vishnu for her critical comments. We acknowledge use of the IISER TVM School of Biology imaging facility to generate microscopy images. We acknowledge the aid of Grammarly, Google Docs and ChatGPT for spelling corrections, grammar corrections and minor refining of parts of the text. The authors subsequently reviewed and edited the content as necessary and take full responsibility for the publication's final content.

#### Competing interests
The authors declare no competing or financial interests.

#### Author contributions
Conceptualization: J.F., J.V.; Data curation: J.F., M.N., J.V.; Formal analysis: J.F., M.N.; Investigation: J.F., A.M.M.P., M.N.; Methodology: J.F., M.N., A.M.M.P.; Project administration: J.V.; Resources: J.V.; Supervision: J.V.; Validation: J.F., M.N.; Visualization: J.F., M.N.; Writing – original draft: J.F., J.V.; Writing – review & editing: J.V.

#### Funding
This work was supported by intramural funds from IISER Thiruvananthapuram to J.V.; a Ramanujan Fellowship to J.V. from the Science and Engineering Research Board (SERB), Department of Science and Technology (DST) (SR/S2/RJN-140/ 2011); an Extra Mural Research Grant from the SERB, DST (EMR/2016/004978); a Core Research Grant from the SERB, DST (CRG/2023/002329); and a Research Grant under Scheme for Transformational and Advances Research in Sciences (STARS) from the Indian Institute of Science (IISc), Ministry of Education, India (MoE) (MoE-STARS/STARS-2/2023-0108). We also thank The University Grants Commission (UGC), India for providing Junior and Senior Research Fellowships to J.F., and the Council of Scientific and Industrial Research (CSIR), India for providing a Junior Research Fellowship to M.N. Open Access funding provided by Indian Institute of Science Education and Research Thiruvananthapuram (IISER TVM). Deposited in PMC for immediate release.

#### Data and resource availability
The raw data used for the manuscript is available at https://doi.org/10.6084/m9. figshare.31007389. All other relevant data and details of resources can be found within the article and its supplementary information.

**Diversity and inclusion statement**

This work involves researchers of different genders and stages of career. It was conducted in an environment that fully supports inclusivity and diversity in science.

**Peer review history**

The peer review history is available online at https://journals.biologists.com/dev/lookup/doi/10.1242/dev.205280.reviewer-comments.pdf

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
