## [Peer Review File · Development (Cambridge, England)]

miR-184 modulates dilp8 to control developmental timing during normal growth conditions and in response to developmental perturbations

Jervis Fernandes, Muhammed Naseem, Ayisha Marwa and Jishy Varghese
DOI: 10.1242/dev.205280

Editor: Thomas Lecuit

Review timeline

Submission to Review Commons:	07 January 2025
Submission to Development:	29 September 2025
Editorial Decision:	19 November 2025
First revision received:	7 January 2026
Accepted:	28 January 2026

Reviewer 1

Evidence, reproducibility and clarity

Summary

In this study, Fernandes and colleagues addressed the question of the role of micro-RNAs in regulating the coupling between organ growth and developmental timing. Using *Drosophila*, they identified the conserved micro-RNA miR-184 as a regulator of the developmental transition between juvenile larval stages and metamorphosis. This transition is under the control of the steroid hormone Ecdysone, and has been shown to be modulated in case of abnormal tissue growth to adjust the duration of larval growth in response to developmental perturbations. The relaxin-like hormone Dilp8 has been identified as a key secreted factor involved in this coupling. Here, the authors show that miR-184 is involved in the regulation of Dilp8 expression both in physiological conditions and upon growth perturbation. They propose that this function is carried out in imaginal tissues, where miR-184 levels are modulated by tissue stress. While several factors have already been involved in triggering sharp dilp8 induction at the transcriptional level, this study adds another level of complexity to the regulation of Dilp8 by proposing that its expression is fine-tuned post-transcriptionally through repression by miR-184.

Major Comments

Overall, the manuscript is well organized, and the logics of the experimental plan well presented. The results are clear, and I appreciate the quality of the pupariation curves. However, I believe that two main conclusions of the paper are not fully supported by the results presented in the figures: the direct regulation of dilp8 3'UTR by miR-184, and the specificity of this regulation in imaginal discs. Here I develop in more details these two aspects.

1. The strategy of the 3'UTR sensor is not fully optimized. Indeed, in most experiments, qRT-PCR is used to assess dilp8 expression levels, although it reflects both transcriptional and post-transcriptional. Importantly, to show that post-transcriptional regulation is involved in the response to tissue damage, the levels of the 3'UTR sensor should be analyzed in discs expressing RAcS (showing at the same time that the response is cell-autonomous in the discs). The expected upregulation of the sensor should be prevented by simultaneous expression of miR-184. This approach would shed light on the relative contribution of transcriptional versus post-

transcriptional regulation of *dilp8* in response to growth perturbation.

2. In my opinion, the use of a 3'UTR sensor is not sufficient to conclude that the regulation by miR-184 is direct, as miR-184 could also regulate an intermediate factor that acts on *dilp8* post-transcriptional regulation. To solve this issue, a common strategy is to generate a 3'UTR sensor with mutated binding sites that should abolish the regulation by miR-184. This mutated 3'UTR might also respond differently to tissue damage, which would strongly support the conclusions of the study.

3. Concerning the tissue-specific regulation of *Dilp8* by miR-184, these results need to be strengthened. Indeed, this comes mostly from phenotypes observed with *rn-GAL4*. Although this is a classical tool for driving expression in imaginal discs, *rn-GAL4* also drives strong expression in other tissues that could contribute to triggering a delay, such as the CNS and part of the gut (proventriculus). In our hands, some growth phenotypes in the wing obtained with *rn-GAL4* could be fully reverted by blocking GAL4 in the CNS indicating that the phenotype was not wing-specific. Importantly, miR-184 seems to be highly expressed in the CNS according to FlyBase, reinforcing the possibility that it plays a role in this organ. Here I propose approaches to confirm that miR-184 mediated regulation of *dilp8* and developmental timing indeed occur in the discs:

- Another driver with less secondary expression sites could be used (*pdmR11F02-GAL4*), or *rn-GAL4* could be combined with an *elav-GAL80* to prevent expression in most neurons.
- The authors could identify the source of *Dilp8* upregulation in miR-184 mutants using tissue-specific qRT-PCR instead of whole larvae expression like in Fig 4A-B.
- This tissue-specific upregulation could be functionally tested using a rescue experiment, in which the delay observed in miR-184 mutants could be rescued by disc-specific downregulation of *Dilp8* (using *pdm2-GAL4* for instance).

Optional: Because it is known that *dilp8* is strongly regulated at the transcriptional level, the relative input from post-transcriptional upregulation is an important question arising from this study. Although it might be a more long-term approach, I believe that generating a *Dilp8* mutant lacking its 3'UTR or, even better, with mutated miR-184 binding sites, would shed light on the role of this regulation for the response to growth perturbation and/or developmental stability (fluctuating asymmetry).

Minor Comments

- I think that a number of results could be moved to SI as they are either controls, or reproduce published data without bringing novelty. For instance, results in Fig 5A-D are similar to data published by Sanchez et al, as stated in the text. Fig6A as well.
- Fig 6D is quite mysterious, as it suggests that basal JNK activation regulates miR-184, which is different from a context of tissue damage. I think that this result could be removed. Alternatively, if the authors want to dig in that direction, more experiments should be provided, such as *bskDN* expression in an RAcS context and the effects on miR-184 levels and the 3'UTR sensor (since transcript levels are already published).
- The references related to *Dilp8* should be checked more in detail in the intro and discussion. About *Dilp8* and developmental stability: remove the ref to Colombani et al 2012, instead put Boone et al 2016 and add Blanco-Obregon et al 2022 (in addition to Garelli et al 2012 who initially identified this phenotype. About *Lgr3* as the receptor for *Dilp8*: add Colombani et al, Current Biology 2015, and cite here Vallejo et al 2015, Garelli et al 2015.
- Among the important transcriptional regulators of *Dilp8*, *Xrp1* could be mentioned (Boulan et al 2019, Destefanis et al 2022) as it plays a complementary function to JNK depending on the type of tissue stress.

Significance

General Assessment

This study provides convincing data showing that the conserved microRNA miR-184 plays a role in regulating developmental timing in *Drosophila* through modulating the levels of *Dilp8*, a key factor in the coupling between tissue growth and developmental transitions. The results are convincing, but the general conclusions of the paper need to be strengthened regarding the direct regulation of *dilp8* by miR-184 and the tissue-specificity of this interaction.

****Advance****

Dilp8 is a key factor that modulates growth and timing in response to developmental perturbations and contributes to developmental precision in physiological conditions. As such, its regulation has been studied by different groups in the last decade, leading to the identification of several inputs for its transcriptional regulation. Here, the authors uncover a post-transcriptional regulation by miR-184, adding another level of regulation of Dilp8 that contribute to ensuring proper regulation of developmental timing, and opening the possibility that miR-184 might play similar roles in other species.

****Audience****

This study is of interest for researchers in the field of basic science, with a focus on developmental timing, tissue damage and biological function of microRNAs.

****Reviewer expertise****

Drosophila, growth control, developmental timing, Dilp8.

Reviewer 2**Evidence, reproducibility and clarity**

Drosophila has helped to characterize the mechanisms that coordinate tissue growth with developmental timing. The insulin/relaxin-like peptide Dilp8 has been identified as a key factor that communicates the abnormal growth status of larval imaginal discs to neuroendocrine neurons responsible for regulating the timing of metamorphosis. Dilp8, derived from imaginal discs, targets four Lgr3-positive neurons in the central nervous system, activating cyclic-AMP signaling in an Lgr3-dependent manner. This signaling pathway reduces the production of the molting hormone, ecdysone, delaying the onset of metamorphosis. Simultaneously, the growth rates of healthy imaginal tissues slow down, enabling the development of proportionate individuals.

In this manuscript "miR-184 modulates dilp8 to control developmental timing during normal growth conditions and in response to developmental perturbations" by Dr. Varghese and colleagues, the authors identify a new post transcriptional regulator of Dilp8. The authors show that miR-184 plays a pivotal role in tissue damage responses by inducing dilp8 expression, which in turn delays pupariation to allow sufficient time for damage repair mechanisms to take effect.

****Major points:****

- In most of the experiments for percentage of pupariation, the 50% pupariation in control is around 110 hours AED in figures 1, 2 and 3. In figures 5 and 6 using the UAS Ricin, the controls are more around 90 hours AED. Why this discrepancy?
- What is the mechanism behind the expression of miR-184 in stress conditions? Does miR-184 also implicated in other conditions giving rise to a developmental delay (X-rays irradiation or animal bearing rasV12, scrib-/- tumors)?
- dilp8 mutant animals have also been shown to be more resistant to starvation or desiccation (<https://doi.org/10.3389/fendo.2020.00461>). Is miR-184 implicated in this answer?
- dilp8 expression has been also shown to be regulated by Xrp1 in response to ribosome stress (<https://doi.org/10.1016/j.devcel.2019.03.016>). This paper should be included in the manuscript Is it possible that the expression levels of miR184 are regulated by Xrp1?

****Minor points:****

- Does the overexpression of miR184 induce an increased fluctuating asymmetry?
- There are 2 references Colombani et al. (2012 for Dilp8 and 2015 for Lgr3). Can you double check that they are used accordingly

Significance

Altogether, the paper present compiling lines of evidence supporting the proposed model. The experiments are well designed and are convincing. The papers is interesting and relevant for a broad audience.

Reviewer 3

Evidence, reproducibility and clarity

This is an interesting study demonstrating an interaction between miR-184 and the *Drosophila* insulin-like peptide 8 (*dilp8*) in the tissue damage response. The authors show that *Dilp8* activity is negatively regulated by miR-184, apparently through direct interaction between miR-184 and the *dilp8*-3'UTR, which leads to lower *dilp8* mRNA transcript levels, via an undetermined mechanism, supposedly its degradation? Furthermore, the authors show that during aberrant tissue growth, miR-184 levels are very slightly downregulated (see comment below), and based on other experiments, imply causation of this with the increased *dilp8* mRNA levels that occur in these tissues, again via an unclear mechanism: upregulation or stabilization of *dilp8* mRNA. The authors present evidence that the JNK pathway, which had been known to be critical for *dilp8* mRNA upregulation upon tissue damage, does so via miR-184.

The data showing the direct regulation of *dilp8*-3'UTR by miR-184 are not very strong and would require more controls to strengthen the claim, as described below. The miR-184 effects are also very small (less than 2-fold reduction with tissue damage; or less than 2-fold induction with JNK-pathway inhibition via *bsk*-DN). These two points are the weakest part of the manuscript and model. Regarding the expression levels, it does not help that the authors show bar graphs with standard errors of the mean instead of the actual datapoints to allow reliable appreciation of the data dispersion. It is difficult to understand how minute changes in miR-184 levels can lead to over an order of magnitude differences (in some cases) in *dilp8* mRNA levels considering that it is a stoichiometric relationship. Maybe miR-184-Dicer1? complexes are highly stable and re-used for multiple *dilp8* transcripts - the authors could discuss how they understand this occurring in their manuscript.

On the same line, discussion is also rather weak on what regards the mechanism of control of *dilp8* mRNA levels by miR-184. Please discuss eg, the evidence for mRNA degradation induction by microRNAs with this UTR binding profile (imperfect UTR binding Fig S4) and-if appropriate-how other possible regulatory models (direct and indirect) could explain the findings.

We suggest the authors carefully revise their citations to cite appropriate work that supports the claims, and also to avoid missing the seminal studies that report the claims they cite.

We have the suggestions below which we hope will help the authors improve their manuscript. If the authors address these points raised above, we believe the manuscript should be a valuable contribution to the field, and help in the understanding of how tissues respond to growth aberrations and the regulation of transcript levels by microRNAs.

Comments:

Results 1st paragraph: please describe the screen in more detail. As written, one only discovers it was a miRNA loss-of-function screen when reading the legend of Table S1. Please show the original data of the screen - with dispersion if possible.

Results 1st paragraph, Fourth line, "While several miRNAs caused delays in pupariation by 12 hours or more..". Please correct, as actually loss of miRNAs caused delays.

Results (Figure 1) - It says that data from three independent experiments are shown. However there is no dispersion in the data. Could the authors please explain this? Are the results of the three experiments summed and presented as one? or is this one of the three?

It is reported in the legend of Figure S2 that LogRank test was performed to determine statistical significance. However, no statistical data is presented. Please show the results.

Fig2A and B. Please show the data points in the bar graphs (as in Figure. 2C), or choose another

data representation. Please consider redoing statistical analysis with a simple t-test. It is not clear to me why ANOVA was used to compare two samples. Please state that data are normalized also to control (tub-GAL4>UAS-scramble). Please state the h post-hatching from which the RNA samples were collected (as in Fig 2C for 20HE quantification).

Fig2C. Fig legend states the bar graphs are "absolute values". Please specify if the bar represents the average, median or something else.

Throughout the manuscript: please use GAL4 in capital letters or at least standardize it throughout the ms. Currently there are GAL4s and Gal4s.. eg compare Fig 2 and 3 legends.

FigS3A and B. Please revise as Fig2A and B above. and apply the same criteria in the respective figure legend.

Fig. 4 - please indicate on the figures what is whole larvae and what is wing imaginal discs. This will facilitate understanding of the figure.

Fig 4 - Data - Authors do not show that rn-GAL4>miR-184-sponge causes up regulation of dilp8 mRNA levels, hence the model is weakened. Doing this experiment would significantly strengthen the study whatever the result is.

The dilp8-3'UTR experiment is weak especially because its generation is not sufficiently well described in the manuscript. "The dilp8 3'UTR-GFP reporter line was created as described in (Vargheese & Cohen, 2007)" is not sufficient. Please describe the construct generation in sufficient detail so that the experiments can be reproduced by others.

Making assumptions, if the construct is as described in Vargheese & Cohen, 2007 and contains all of the dilp8 3'UTR - it should be a Tubulin-driven GFP gene with a dilp8-3'UTR "Tub-GFP-(dilp8 3'UTR)". In this case the authors need to rule out the alternative interpretation of the result in Fig. 4D by showing that the expression of miR-184 does not down regulate Tub-GFP expression itself. The best scenario would be to have a mutated dilp8 3'UTR for the miR-184 recognition site. This experiment would significantly strengthen the study and model.

Figure 4C-D please separate dilp8 from 3'UTR with a space or hyphen.

Figure 4E. Please name the dilp8 allele as MI00727 as it is not a KO, but rather a hypomorphic mutation (fully WT dilp8 transcripts are still generated, albeit at a much lower level).

Figure 6D: please add UAS to bskDN/+. All figures have rn-GAL4 alone or with UAS-GFP as control. This finding would be strengthened with this other control, especially because the size effect is small. This being said a general comment for all experiments is that hemi-controls are generally missing for all figures. eg, in Fig 3. One would typically include controls such as A. Phm>+ and +>miR.184; B. aug21>+ and +>miR.184; C. ptth>+ and +>miR.184; D. rn>+ and +>miR.184

Figure 7: Are IPCs necessary for the model? If not, I suggest removing them and placing the Lgr3 neuron cell bodies much more anterior in this scheme. Their cell bodies are as anterior and rostral as it gets, approximately where the IPCs are depicted in this type of view of the CNS.

Table S1- It would be preferable to see the data of these experiments, but if the authors prefer to show this data in a table, please at least add the dispersion analyses (eg standard deviation.. OR median+-quartiles OR Confidence intervals..), N of animals analysed, and statistics against controls.

In all figures with pupariation time: please also indicate significant findings in the graphs (with an asterisk, for instance) and adjust figure legends accordingly. This could facilitate understanding the data.

Please revise Figure legends for punctuation.

Abstract:

Line 10: What is the evidence to call Dilp8 a "paracrine" factor?

Introduction:

4th paragraph, 3rd sentence " Dilp8... buffers developmental noise and delays pupariation..."
Buffering of developmental noise was first shown in Garelli et al., Science 2012, so this publication should be cited.

4th paragraph, 5th sentence: please include Jaszczak et al., Genetics 2016. This paper was published together with the 2015 papers, just a matter of timing that it got a 2016 date. Moreover, I do not think Katsuyama et al., 2015 is well cited to back up the statement in this sentence, hence I recommend removing that citation in this sentence.

6th paragraph: 5th line "targeting dilp8" : please specify if you mean the gene or the mRNA, or both.

Same for line 7.

Results Page 10, 1st paragraph, 1st sentence: the works cited are not the appropriate studies that demonstrated what is being stated. This was shown in Garelli et al., Science 2012 and Colombani et al., Science 2012.

Results Page 10, 1st paragraph, line 11: Please also cite Colombani et al., Science 2012, who first showed that JNK is required for dilp8 regulation.

Discussion, 2nd paragraph, line 4: again, please indicate the rationale for using "paracrine" to describe Dilp8's activities. The current widely accepted model is that Dilp8 acts on interneurons in the brain (eg, reviewed in Juarez-Carreño et al., Cell Stress, 2018; Gontijo and Garelli, Mech Dev, 2018; Mirth and Shingleton, Front Cell Dev Biol, 2019; Texada et al., Genetics 2020; Boulan and Leopold, 2021). In order to reach the brain, Dilp8 has to be secreted from the discs and travel to the brain. This is as an endocrine mechanism as it gets for a small larva, considering that some discs can be in the opposite side of the larva (eg, genital discs). While this does not exclude that Dilp8 could also act paracrinely, the only evidence that I am aware of comes from other contexts such as during transdetermination (where Dilp8 has been proposed to work in an autocrine or paracrine fashion, via Drl in imaginal discs (Nemoto et al., Genes to Cells, 2023), however, this is not cited appropriately in this manuscript and is less related to the Lgr3-dependent pathway being studied here.

Discussion Page 13, 1st paragraph, This claim is supported by data presented in Garelli et al., Science 2012, not the other two papers. Garelli et al., 2015 shows that the Lgr3 receptor also participates in buffering developmental noise. Other studies have corroborated the Garelli et al., 2012 finding: eg, Colombani et al., Curr Biol 2015; Boone et al., Nat Commun 2016; Blanco-Obregon et al., Nat Commun 2022). Many other studies have shown that Dilp8 promotes developmental stability under tissue stress and challenges.

Discussion Page 12, 3rd paragraph, 2nd sentence: "The Lgr3 neurons directly interact with ... PTHH ...and insulin-producing neurons" Please cite Colombani et al., 2015 and Vallejo et al., Science 2015. Vallejo et al., propose that circuit with insulin-producing neurons. In the 3rd sentence, only Jaszczak et al., 2016 is cited, whereas this claim/model comes from many studies, such as Halme et al., Curr Biol, 2010; Hackney et al., PLoS One 2012; Garelli et al. Science 2012; Colombani et al., Science, 2012; and the Lgr3 papers from 2015).
Jaszczak et al., actually propose that Lgr3 is also required in the ring gland in addition to neurons.

Discussion page 14 last paragraph, 10 line, "In *Aedes aegypti* regulates ilp8 (Ling et al., 2017)". As far as I understand mosquitoes do not have a dilp8 orthologue (see for instance Gontijo and Gontijo, Mech Dev 2018; and Jan Veenstra's work). ilp nomenclature (numbering) does not follow that of *Drosophila*, so ilp8 is probably a typical Insulin/IGF-like peptide and is NOT an orthologue of Dilp8, a relaxin, so this citation needs to be removed or placed into the broader context of microRNA regulation of ilps.

Thank you for the opportunity to review your interesting work,
Alisson Gontijo and Rebeca Zanini

Significance

If the authors address these points raised above, we believe the manuscript should be a valuable contribution to the field, and help in the understanding of how tissues respond to growth aberrations and the regulation of transcript levels by microRNAs.

Author response to reviewers' comments

1. General Statements

We thank the reviewers for your positive responses regarding our manuscript. The reviewers felt that our work is interesting and convincing, they suggested experiments to strengthen the manuscript further, other revision suggestions on improving the text and corrections in citations were also made. We have performed new experiments and incorporated all the suggestions of the reviewers. We believe these changes have significantly improved our manuscript. We trust that this work will be of interest to biologists studying developmental processes and responses to developmental defects. Our findings have broader implications for understanding how organisms, including humans, buffer the impacts of injuries and infections during early life stages. We have included a 'revised manuscript' and 'revised manuscript-with all changes tracked' versions with the resubmission.

Reviewer #1

Evidence, reproducibility and clarity

SUMMARY

In this study, Fernandes and colleagues addressed the question of the role of micro-RNAs in regulating the coupling between organ growth and developmental timing. Using *Drosophila*, they identified the conserved micro-RNA miR-184 as a regulator of the developmental transition between juvenile larval stages and metamorphosis. This transition is under the control of the steroid hormone Ecdysone, and has been shown to be modulated in case of abnormal tissue growth to adjust the duration of larval growth in response to developmental perturbations. The relaxin-like hormone Dilp8 has been identified as a key secreted factor involved in this coupling. Here, the authors show that miR-184 is involved in the regulation of Dilp8 expression both in physiological conditions and upon growth perturbation. They propose that this function is carried out in imaginal tissues, where miR-184 levels are modulated by tissue stress. While several factors have already been involved in triggering sharp dilp8 induction at the transcriptional level, this study adds another level of complexity to the regulation of Dilp8 by proposing that its expression is fine-tuned post-transcriptionally through repression by miR-184.

MAJOR COMMENTS

Overall, the manuscript is well organized, and the logics of the experimental plan well presented. The results are clear, and I appreciate the quality of the pupariation curves. However, I believe that two main conclusions of the paper are not fully supported by the results presented in the figures: the direct regulation of dilp8 3'UTR by miR-184, and the specificity of this regulation in imaginal discs. Here I develop in more details these two aspects.

Comment 1) The strategy of the 3'UTR sensor is not fully optimized. Indeed, in most experiments, qRT-PCR is used to assess *dilp8* expression levels, although it reflects both transcriptional and post-transcriptional. Importantly, to show that post-transcriptional regulation is involved in the response to tissue damage, the levels of the 3'UTR sensor should be analyzed in discs expressing RAcS (showing at the same time that the response is cell-autonomous in the discs). The expected upregulation of the sensor should be prevented by simultaneous expression of *miR-184*. This approach would shed light on the relative contribution of transcriptional versus post-transcriptional regulation of *dilp8* in response to growth perturbation.

Response: We thank the reviewer for this comment. We agree that qRT-PCRs do not distinguish between transcriptional and post-transcriptional changes of *dilp8* levels, in response to changes in *miR-184* levels and tissue damage. In addition to the qRT-PCR data we have looked at *dilp8*-3'UTR-GFP reporter in response to overexpression of *miR-184* in the wingdisc using *patched*-Gal4 driver, which show downregulation of the GFP reporter in the *ptc* domain (Fig 4C-D'). This suggests that *dilp8* mRNA is a direct target of *miR-184* by post-transcriptional regulation through its 3'UTR. Further, to confirm the specificity of the effect of *miR-184* on *dilp8*-3'UTR, we generated a *dilp8*-3'UTR mutant in which the single target site for *miR-184* was mutated. We show that the mutated *dilp8*-3'UTR reporter doesn't show any regulation in response to *miR-184* overexpression in the *ptc* domain of the wingdisc (Fig. 4E, E', F, F'). This experiment confirms the specificity of the *dilp8*-3'UTR regulation by *miR-184*.

As suggested by the reviewer we analysed *dilp8*-3'UTR-GFP reporter expression by overexpressing RicinA using *ptc*GAL4 driver in the wing imaginal disc (Fig. S6F-G'). We observed a slight but consistent increase in the *dilp8*-3'UTR-GFP reporter expression, indicating post-transcriptional regulation of *dilp8* expression in response to tissue damage. However, the increase of reporter GFP levels observed in this experiment in response to tissue damage is mild (Fig. S6F-G') than expected based on the qRT-PCR results (Fig S6A and B). We have added this new data to the manuscript (Fig. S6F-G').

We propose the following reasons to explain this result:

- a) both transcriptional and post-transcriptional regulation of *dilp8* mRNA in response to developmental perturbations
- b) the data on 3'UTR reporter GFP is specifically from the *ptc* domain expression of RicinA^{CS}, whereas for *dilp8* transcript levels we have expressed RicinA^{CS} in all larval imaginal tissues, or in the entire wing imaginal disc, which could be one of the reasons for the stronger effect seen on *dilp8* mRNA levels
- c) we are not certain if the *tubulin*-promoter driven *dilp8*-3'UTR GFP reporter reflects post-transcriptional regulation of *dilp8* by *miR-184* efficiently in comparison to qRT-PCR. This is especially as the reporter-GFP-3'UTR will be expressed at very high levels due to the *tubulin* promoter, a majority of this reporter-GFP mRNA may not be relieved from degradation due to the moderate suppression of *miR-184* in response to RicinA^{CS} overexpression.

Thus, our experiments suggest that *dilp8* levels are regulated post-transcriptionally by *miR-184* which contributes to pupariation delays in response to tissue damage. In support of this, we could rescue pupariation delays and *dilp8* induction caused by RicinA^{CS} expression using overexpression of *miR-184* (Figs 5B, C). Thus, we confirm that the effect of post-transcriptional regulation by *miR-184* during developmental perturbations also contributes to *dilp8* induction and pupariation delays. Unfortunately, due to experimental limitations we could not perform simultaneous expression of RicinA^{CS} and *miR-184* to evaluate the rescue of *dilp8*-3'UTR-GFP sensor expression. The levels of *dilp8*-3'UTR sensor GFP is reduced efficiently by *miR-184* overexpression (Fig 4D), which prevented us from attempting the rescue of the moderate increase of *dilp8*-3'UTR GFP levels in response to RicinA^{CS}.

Comment 2) In my opinion, the use of a 3'UTR sensor is not sufficient to conclude that the regulation by miR-184 is direct, as miR-184 could also regulate an intermediate factor that acts on *dilp8* post-transcriptional regulation. To solve this issue, a common strategy is to generate a 3'UTR sensor with mutated binding sites that should abolish the regulation by miR-184. This mutated 3'UTR might also respond differently to tissue damage, which would strongly support the conclusions of the study.

Response: We couldn't agree more with the reviewer, this comment is addressed in the response to comment 1. We have confirmed the specificity of regulation of *dilp8*-3'UTR by miR-184 using target site mutated *dilp8*-3'UTR (new figures added to the manuscript Fig. 4E, E', F, F'). We tested if the changes in *dilp8* mRNA levels in response to tissue damage is post-transcriptional mediated by *miR-184*. We observe that there is a slight, but consistent increase of *dilp8*-3'UTR GFP reporter levels in the *ptc* domain of wingdisc in response to RicinA^{CS} expression, suggesting a role for miR-184 mediated post-translational regulation of *dilp8*. However, we have not yet tested the mutated *dilp8*-3'UTR GFP reporter in response to tissue damage.

Comment 3) Concerning the tissue-specific regulation of Dilp8 by miR-184, these results need to be strengthened. Indeed, this comes mostly from phenotypes observed with *rn-GAL4*. Although this is a classical tool for driving expression in imaginal discs, *rn-GAL4* also drives strong expression in other tissues that could contribute to triggering a delay, such as the CNS and part of the gut (proventriculus). In our hands, some growth phenotypes in the wing obtained with *rn-GAL4* could be fully reverted by blocking GAL4 in the CNS indicating that the phenotype was not wing-specific. Importantly, miR-184 seems to be highly expressed in the CNS according to FlyBase, reinforcing the possibility that it plays a role in this organ. Here I propose approaches to confirm that miR-184 mediated regulation of *dilp8* and developmental timing indeed occur in the discs:

- Another driver with less secondary expression sites could be used (*pdmR11F02-GAL4*), or *rn-GAL4* could be combined with an *elav-GAL80* to prevent expression in most neurons.
- The authors could identify the source of Dilp8 upregulation in miR-184 mutants using tissue-specific qRT-PCR instead of whole larvae expression like in Fig 4A-B.
- This tissue-specific upregulation could be functionally tested using a rescue experiment, in which the delay observed in miR-184 mutants could be rescued by disc-specific downregulation of Dilp8 (using *pdm2-GAL4* for instance).

Response: We are thankful to the reviewer, and agree that it is important to show that the effects that we see using *rn-Gal4* are specific to imaginal discs, and not due to an effect in CNS. We tested this by expressing miR-184 sponge in the CNS. Though miR-184 is highly expressed in the larval CNS, downregulation of miR-184 specifically in the pan-neuronal background using *elav-GAL4* led to no effects on pupariation timepoint. We have added this as supplementary data Figure S4. Therefore, we believe that the *miR-184* downregulation phenotype in the *rnGAL4* background can be mainly attributed to its role in the imaginal discs. In addition, as suggested by the reviewer we have also demonstrated that downregulation of miR-184 in the imaginal discs using *rnGAL4* driver leads to an increase in *dilp8* expression (Fig S5B). Thus confirming that *dilp8* mRNA is enhanced in the imaginal discs by blocking *miR-184*.

OPTIONAL: Because it is known that *dilp8* is strongly regulated at the transcriptional level, the relative input from post-transcriptional upregulation is an important question arising from this study. Although it might be a more long-term approach, I believe that generating a Dilp8 mutant lacking its 3'UTR or, even better, with mutated miR-184 binding sites, would shed light on the role of this regulation for the response to growth perturbation and/or developmental stability (fluctuating asymmetry).

Response: We thank the reviewer for the suggestion. This would have been an interesting experiment to carry out especially in the context of fluctuating asymmetry.

MINOR COMMENTS

1. I think that a number of results could be moved to SI as they are either controls, or

reproduce published data without bringing novelty. For instance, results in Fig 5A-D are similar to data published by Sanchez et al, as stated in the text. Fig6A as well.

Response: We thank the reviewer for this suggestion, Fig. 5A-D, and F has been moved to Fig. S6A-E. We have also moved data from Fig. 6 to Fig. 5, as a result Fig 6 A-D has become Fig. 5 B-D.

2. Fig 6D is quite mysterious, as it suggests that basal JNK activation regulates miR-184, which is different from a context of tissue damage. I think that this result could be removed. Alternatively, if the authors want to dig in that direction, more experiments should be provided, such as *bskDN* expression in an RACs context and the effects on miR-184 levels and the 3'UTR sensor (since transcript levels are already published).

Response: We would like to clarify that our experiments suggest that endogenous JNK signalling negatively regulates *miR-184*, as blocking basal JNK signalling using *bskDN* increased the levels of *miR-184* (changed to Fig 5D). Enhanced JNK signalling has been reported to be involved in tissue damage responses, and we propose that *RicinA^{CS}* mediated increase in JNK signalling leads to the reduction of *miR-184* (changed to Fig 5A, S6D-E). However, we are not strongly implying this as we did not co-express *RicinA^{CS}* and *bskDN* to show that JNK signalling is responsible for the drop in miR-184 levels in response to tissue damage. We thank the reviewer for seeking this explanation, we have rewritten the results section to improve clarity.

3. The references related to *Dilp8* should be checked more in detail in the intro and discussion. About *Dilp8* and developmental stability: remove the ref to Colombani et al 2012, instead put Boone et al 2016 and add Blanco-Obregon et al 2022 (in addition to Garelli et al 2012 who initially identified this phenotype. About *Lgr3* as the receptor for *Dilp8*: add Colombani et al, Current Biology 2015, and cite here Vallejo et al 2015, Garelli et al 2015. Among the important transcriptional regulators of *Dilp8*, *Xrp1* could be mentioned (Boulan et al 2019, Destefanis et al 2022) as it plays a complementary function to JNK depending on the type of tissue stress.

Response: We are really sorry for the glaring errors in citing appropriate references. We thank the reviewer for correcting this for us. We have made necessary changes to the text.

Significance

GENERAL ASSESSMENT

This study provides convincing data showing that the conserved microRNA miR-184 plays a role in regulating developmental timing in *Drosophila* through modulating the levels of *Dilp8*, a key factor in the coupling between tissue growth and developmental transitions. The results are convincing, but the general conclusions of the paper need to be strengthened regarding the direct regulation of *dilp8* by miR-184 and the tissue-specificity of this interaction.

ADVANCE

Dilp8 is a key factor that modulates growth and timing in response to developmental perturbations and contributes to developmental precision in physiological conditions. As such, its regulation has been studied by different groups in the last decade, leading to the identification of several inputs for its transcriptional regulation. Here, the authors uncover a post-transcriptional regulation by miR-184, adding another level of regulation of *Dilp8* that contribute to ensuring proper regulation of developmental timing, and opening the possibility that miR-184 might play similar roles in other species.

AUDIENCE

This study is of interest for researchers in the field of basic science, with a focus on developmental timing, tissue damage and biological function of microRNAs.

REVIEWER EXPERTISE

Drosophila, growth control, developmental timing, *Dilp8*.

Reviewer #2**Evidence, reproducibility and clarity**

Drosophila has helped to characterize the mechanisms that coordinate tissue growth with developmental timing. The insulin/relaxin-like peptide *Dilp8* has been identified as a key factor that communicates the abnormal growth status of larval imaginal discs to neuroendocrine neurons responsible for regulating the timing of metamorphosis. *Dilp8*, derived from imaginal discs, targets four *Lgr3*-positive neurons in the central nervous system, activating cyclic-AMP signaling in an *Lgr3*-dependent manner. This signaling pathway reduces the production of the molting hormone, ecdysone, delaying the onset of metamorphosis. Simultaneously, the growth rates of healthy imaginal tissues slow down, enabling the development of proportionate individuals.

In this manuscript "miR-184 modulates *dilp8* to control developmental timing during normal growth conditions and in response to developmental perturbations" by Dr. Varghese and colleagues, the authors identify a new post transcriptional regulator of *Dilp8*. The authors show that miR-184 plays a pivotal role in tissue damage responses by inducing *dilp8* expression, which in turn delays pupariation to allow sufficient time for damage repair mechanisms to take effect.

Major points:

Comment 1) In most of the experiments for percentage of pupariation, the 50% pupariation in control is around 110 hours AED in figures 1, 2 and 3. In figures 5 and 6 using the UAS Ricin, the controls are more around 90 hours AED. Why this discrepancy?

Response: We thank the reviewer for asking for this clarification. The former experiments for Figs 1-3 were carried out at 25°C while the latter experiments with a cold sensitive version of *RicinA* (*UAS-RA^{CS}*), Figs 5 and 6 (now changed to Figs. 5 and S6 as suggested by reviewer #1) were carried out at 29°C (permissive temperature). This difference in temperature has led to alterations in pupariation timing. We apologise for not having mentioned this in the text, now we have made necessary corrections to the methods section clearly indicating this.

Comment 2) What is the mechanism behind the expression of miR-184 in stress conditions? Is miR-184 also implicated in other conditions giving rise to a developmental delay (X-rays irradiation or animal bearing *rasV12*, *scrib*^{-/-} tumors)?

Response: We thank the reviewer for these questions.

a) In response to developmental perturbations by *RicinA*, we believe that activation of JNK signalling controls miR-184 expression. We propose this as our experiments show that imaginal disc damage leads to enhancement of JNK signalling and increase in *dilp8* mRNA levels (as reported earlier by Colombani et al 2012; Sánchez et al 2019), and a simultaneous reduction of *miR-184* (Figs. S6A, D, E). We also have performed new experiments to show that in response to *RicinA* expression in the wingdisc there is moderate increase in the *dilp8*-3'UTR-GFP sensor expression (Figs. S6F-G'), indicating a post-transcriptional regulation of *dilp8* expression in response to tissue stress. We also show that *RicinA* induced *dilp8* expression and pupariation delay can be rescued by increasing miR-184 levels (Fig 5B and C), suggesting that the reduction of miR-184 in response to tissue damage contributes to the damage responses. In a separate experiment we show that blocking the endogenous JNK pathway by the expression of *bskDN* enhances miR-184 levels, suggesting that miR-184 is under the regulation of JNK signalling (Fig 5D). Hence, we speculate that during tissue stress, activation of JNK signalling leads to a reduction of miR-184 levels which contributes to regulating the levels of *dilp8* post-transcriptionally and resulting in pupariation delays. The text has been modified to explain this better.

b) In a previous paper by Shu et al., 2017 (<https://doi.org/10.18632/oncotarget.22226>) decreased expression of miR-184 was observed in a *IgIRNAi*; *RasV12* tumor background. Apart from this various studies have shown that *dilp8* levels increase in response to tumour, radiation stress,

apoptosis, and tissue damage (Yeom et al 2021, Ray et al 2019, Demay et al 2014, Katsuyama et al 2015, Colombani et al 2012, Garelli et al 2012). Whether the regulation of *dilp8* by miR-184, occurs in these backgrounds is yet to be tested. We have now discussed this possibility in the manuscript.

Comment 3) *dilp8* mutant animals have also been shown to be more resistant to starvation or desiccation (<https://doi.org/10.3389/fendo.2020.00461>). Is miR-184 implicated in this answer?

Response: We thank the reviewer for this question. In our earlier experiments miR-184 has been demonstrated to be regulated by nutrition in the larval stages and lack of miR-184 led to enhanced larval death in response to diet restriction (Fernandes et al., 2022). miR-184 was also demonstrated to play a role in the insulin producing cells (IPCs) in regulating lifespan (Fernandes & Varghese., 2022). In the current work, we propose miR-184 to act upstream of *dilp8* in response to stress stimuli. Hence, it is possible that miR-184 might be involved in responses to starvation and desiccation stress in the adult female flies, by regulating *dilp8* levels post-transcriptionally. However, it has not been tested yet if the miR-184 regulation of *dilp8* plays a role in resistance to starvation or desiccation in adult females, as this was not within the scope of the current study. We have now added this reference in the discussion section.

Comment 4) *dilp8* expression has been also shown to be regulated by Xrp1 in response to ribosome stress (<https://doi.org/10.1016/j.devcel.2019.03.016>). This paper should be included in the manuscript. Is it possible that the expression levels of miR184 are regulated by Xrp1?

Response: We thank the reviewer for the suggestion and have incorporated the reference into the paper. During ribosome stress in the larval imaginal discs the stress-response transcription factor Xrp1 acts through *dilp8* in regulating systemic growth. We agree with the reviewer, it is possible that expression of miR-184 is regulated by Xrp1. Currently we have not explored this possibility. We have now added this to the discussion section.

Minor points:

1. Does the overexpression of miR184 induce an increased fluctuating asymmetry?

Response: We thank the reviewer for asking this question. The role of *dilp8* in the fluctuation asymmetry is only observed in the *dilp8* hypomorphic mutant background. To replicate this we would have to overexpress miR-184 in either the whole larvae or in the wing discs. Unfortunately overexpression of miR-184 in the wing discs (using *rnGAL4*) leads to pupal lethality while as overexpression of miR-184 in the whole larvae leads to embryonic lethality and therefore we were not be able to conclude from our experiments if miR-184 overexpression induces increased fluctuating asymmetry.

2. There are 2 references Colombani et al. (2012 for Dilp8 and 2015 for Lgr3). Can you double check that they are used accordingly

Response: We thank the reviewer for pointing these errors out and we have incorporated these changes into the paper.

Significance

Altogether, the paper present compiling lines of evidence supporting the proposed model. The experiments are well designed and are convincing. The papers is interesting and relevant for a broad audience.

Reviewer #3

Evidence, reproducibility and clarity (Required):

This is an interesting study demonstrating an interaction between miR-184 and the *Drosophila* insulin-like peptide 8 (*dilp8*) in the tissue damage response. The authors show that Dilp8 activity is

negatively regulated by miR-184, apparently through direct interaction between miR-184 and the *dilp8*-3'UTR, which leads to lower *dilp8* mRNA transcript levels, via an undetermined mechanism, supposedly its degradation? Furthermore, the authors show that during aberrant tissue growth, miR-184 levels are very slightly downregulated (see comment below), and based on other experiments, imply causation of this with the increased *dilp8* mRNA levels that occur in these tissues, again via an unclear mechanism: upregulation or stabilization of *dilp8* mRNA. The authors present evidence that the JNK pathway, which had been known to be critical for *dilp8* mRNA upregulation upon tissue damage, does so via miR-184.

Major Comments:

Comment 1: The data showing the direct regulation of *dilp8*-3'UTR by miR-184 are not very strong and would require more controls to strengthen the claim, as described below.

Response: We have performed new experiments to validate that *dilp8*-3'UTR is regulated by miR-184. Please see the detailed responses to comments 10-12 below.

Comment 2: The miR-184 effects are also very small (less than 2-fold reduction with tissue damage; or less than 2-fold induction with JNK-pathway inhibition via *bskDN*). These two points are the weakest part of the manuscript and model.

Response: We agree with the reviewers on this point. The reduction in miR-184 levels in response to RicinA expression is modest (25-30%), and the induction of miR-184 in response to *bskDN* expression is less than two-fold (Figs. 5A and D). In contrast, *dilp8* transcript levels increase several-fold in response to RicinA expression (Fig. 5C, S6A and B). Since we measure *dilp8* transcript levels by qPCR, we detect both transcriptional and post-transcriptional contributions to *dilp8* regulation. In addition, we have performed a new experiment to check the post-transcriptional regulation of *dilp8*, in response to tissue damage. Though the change in the *dilp8*-3'UTR GFP reporter upon RicinA expression in the *ptc* domain of the wingdisc is mild (Figs. S6F-G'), this strongly suggests a post-transcriptional outcome of the reduction of miR-184 levels on *dilp8*. Hence, we propose that tissue damage induces strong transcriptional activation of *dilp8*, while the reduction of miR-184, despite its smaller magnitude, contributes to *dilp8* upregulation via post-transcriptional regulation. In support of this, our experiments demonstrate direct regulation of the *dilp8*-3'UTR by miR-184 (Figs. 4C-F'), and show strong *dilp8* mRNA upregulation in miR-184 deficient conditions (Fig. 4A and B), suggesting the role of miR-184 in maintaining *dilp8* levels. We also show that RicinA induced effects on *dilp8* and pupariation delay are reversed by co-expression of miR-184 (Fig. 5C). We do not claim that regulation by miR-184 is the sole mechanism for driving *dilp8* induction during tissue damage, but suggest that miR-184-mediated post-transcriptional regulation acts in a complementary manner to transcriptional responses. Furthermore, we believe that the mild effect of JNK signaling on miR-184 (as shown by the *bskDN* experiment) is sufficient for the moderate reduction of miR-184 in response to tissue damage.

Comment 3: Regarding the expression levels, it does not help that the authors show bar graphs with standard errors of the mean instead of the actual data points to allow reliable appreciation of the data dispersion.

Response: We have modified our figures and have performed statistical analysis according to the suggestions of the reviewers, please see responses to comments 1-9, and 13-19.

Comment 4: It is difficult to understand how minute changes in miR-184 levels can lead to over an order of magnitude differences (in some cases) in *dilp8* mRNA levels considering that it is a stoichiometric relationship. Maybe ?miR-184-Dicer1? complexes are highly stable and re-used for multiple *dilp8* transcripts - the authors could discuss how they understand this occurring in their manuscript.

On the same line, discussion is also rather weak on what regards the mechanism of control of *dilp8* mRNA levels by miR-184. Please discuss eg, the evidence for mRNA degradation induction by microRNAs with this UTR binding profile (imperfect UTR binding Fig S4) and-if appropriate-how other possible regulatory models (direct and indirect) could explain the findings.

Response: We accept the reviewers comment that 25-30% reduction of miR-184 is low in

comparison to the many fold increase in *dilp8* levels. We believe that both post-transcriptional and transcriptional changes are responsible for the induction of *dilp8* in response to tissue damage. However, our experiments suggest the role of post-transcriptional regulation by miR-184, as pupariation delay is rescued by miR-184 overexpression (also please see the response to the previous comment). We are not ruling out the possibility of transcriptional regulation of *dilp8* mRNA, rather we are suggesting the possibility that both transcriptional and post-transcriptional means are responsible for changes in *dilp8*. Moreover, we have not performed absolute measurement of miR-184 in the imaginal discs (what we show is a comparison between control and RicinA expression), hence we do not have an exact estimate of how many miR-184 molecules are reduced and if they would be greatly equal or more in comparison to the *dilp8* mRNA molecules that are upregulated, as again while measuring *dilp8* mRNA we are not checking how many molecules of *dilp8* exactly are increased. As the reviewer suggests, it is possible that miR-184-RISC could be stable to handle multiple *dilp8* molecules one after the other, hence it is not a 1:1 relationship between miR-184:*dilp8*. We have included this in the manuscript. It is also known that imperfect 3'UTR binding as seen in most animal microRNAs leads to translational repression and mRNA deadenylation, which eventually results in mRNA degradation.

Comment 5: We suggest the authors carefully revise their citations to cite appropriate work that supports the claims, and also to avoid missing the seminal studies that report the claims they cite.

Response: We are really apologetic for the errors citing the key references. We are grateful to the reviewers for correcting this for us. We have made changes to the text to include and correct the references.

We have the suggestions below which we hope will help the authors improve their manuscript. If the authors address these points raised above, we believe the manuscript should be a valuable contribution to the field, and help in the understanding of how tissues respond to growth aberrations and the regulation of transcript levels by microRNAs.

Comments:

Comment 1. Results 1st paragraph: please describe the screen in more detail. As written, one only discovers it was a miRNA loss-of-function screen when reading the legend of Table S1. Please show the original data of the screen - with dispersion if possible.

Response: We thank the reviewers for these suggestions, we have now included the data from the screen with SEM, and p-values.

Comment 2. Results 1st paragraph, Fourth line, "While several miRNAs caused delays in pupariation by 12 hours or more..". Please correct, as actually loss of miRNAs caused delays.

Response: We thank the reviewer for pointing out this error, we have corrected the text accordingly.

Comment 3. Results (Figure 1) - It says that data from three independent experiments are shown. However there is no dispersion in the data. Could the authors please explain this? Are the results of the three experiments summed and presented as one? or is this one of the three?

Response: We thank the reviewers for these suggestions and have plotted data with the SEM values.

Comment 4. It is reported in the legend of Figure S2 that LogRank test was performed to determine statistical significance. However, no statistical data is presented. Please show the results.

Response: We thank the reviewers for these suggestions to improve the data presentation, we have incorporated the p-value as suggested.

Comment 5. Fig2A and B. Please show the data points in the bar graphs (as in Figure. 2C), or choose another data representation. Please consider redoing statistical analysis with a simple t-

test. It is not clear to me why ANOVA was used to compare two samples. Please state that data are normalized also to control (*tub-GAL4>UAS-scramble*). Please state the h post-hatching from which the RNA samples were collected (as in Fig 2C for 20HE quantification).

Response: We thank the reviewers for these suggestions to improve the data presentation, we have incorporated all changes as suggested. Similar changes have been incorporated to the rest of the figures of the manuscript as well. Hours post-hatching information for each figure is now added to the figure legends.

Comment 6. Fig2C. Fig legend states the bar graphs are "absolute values". Please specify if the bar represents the average, median or something else.

Response: We thank the reviewer for pointing this out, we have made the suggested changes.

Comment 7. Throughout the manuscript: please use GAL4 in capital letters or at least standardize it throughout the ms. Currently there are GAL4s and Gal4s.. eg compare Fig 2 and 3 legends.

Response: We thank the reviewer for pointing this out, we have incorporated all changes as recommended.

Comment 8. FigS3A and B. Please revise as Fig2A and B above. and apply the same criteria in the respective figure legend.

Response: We thank the reviewer for pointing this out, we have made the changes as recommended.

Comment 9. Fig. 4 - please indicate on the figures what is whole larvae and what is wing imaginal discs. This will facilitate understanding of the figure.

Response: We thank the reviewers for these suggestions and have included this information in all the figures.

Comment 10. Fig 4 - Data - Authors do not show that *rn-GAL4>miR-184-sponge* causes up regulation of *dilp8* mRNA levels, hence the model is weakened. Doing this experiment would significantly strengthen the study whatever the result is.

Response: We thank the reviewer for pointing this out and we have included this in the manuscript (Fig S5B).

Comment 11. The *dilp8-3'UTR* experiment is weak especially because its generation is not sufficiently well described in the manuscript. "The *dilp8 3'UTR-GFP* reporter line was created as described in (Vargheese & Cohen, 2007)" is not sufficient. Please describe the construct generation in sufficient detail so that the experiments can be reproduced by others.

Response: We thank the reviewer for pointing this out and we have elaborated in the methods section on how we generated the *dilp8 3'UTR-GFP* reporter and *dilp8 3'UTR* mutant *GFP* reporter lines. The plasmid was originally created in Steve Cohen's lab at EMBL, by modifying pCasper4 plasmid, by introducing a *tubulin* promoter, EGFP and a multiple cloning site, which allows one to clone 3'UTRs of target genes into this plasmid. *Not1* and *Xho1* sites were used to clone the *dilp8-3'UTR* and *mut-3'UTR*. We hope this explains our strategy sufficiently.

Comment 12. Making assumptions, if the construct is as described in Vargheese & Cohen, 2007 and contains all of the *dilp8 3'UTR* - it should be a Tubulin-driven GFP gene with a *dilp8-3'UTR* "Tub-GFP-(*dilp8 3'UTR*)". In this case the authors need to rule out the alternative interpretation of the result in Fig. 4D by showing that the expression of miR-184 does not down regulate Tub-GFP expression itself. The best scenario would be to have a mutated *dilp8 3'UTR* for the miR-184 recognition site. This experiment would significantly strengthen the study and model.

Response: We thank the reviewer for pointing this out. We agree with the reviewers that this experiment is needed to prove direct regulation of the *dilp8*-3'UTR by miR-184. We have mutated the sequences complementary to the seed region of miR-184 in the *dilp8*-3'UTR, and demonstrated that overexpression of miR-184 does not regulate the mutated *tub*-GFP-(*dilp8* 3'UTR) expression. This confirms that the *dilp8* gene is a direct target of *miR-184*. This data is added to the manuscript as Figs 4E-F'.

Comment 13. Figure 4C-D please separate *dilp8* from 3'UTR with a space or hyphen.

Response: We thank the reviewer for pointing this out and have separated *dilp8* from 3'UTR with a hyphen.

Comment 14. Figure 4E. Please name the *dilp8* allele as MI00727 as it is not a KO, but rather a hypomorphic mutation (fully WT *dilp8* transcripts are still generated, albeit at a much lower level).

Response: We thank the reviewer for pointing this out and we have made the necessary changes.

Comment 15. Figure 6D: please add UAS to *bskDN/+*. All figures have *rn*-GAL4 alone or with UAS-GFP as control. This finding would be strengthened with this other control, especially because the size effect is small. This being said a general comment for all experiments is that hemi-controls are generally missing for all figures. eg, in Fig 3. One would typically include controls such as A. *Phm>+* and *>miR.184*; B. *aug21>+* and *>miR.184*; C. *ptth>+* and *>miR.184*; D. *rn>+* and *>miR.184*

Response: We thank the reviewer for pointing this out. We have added UAS to *bskDN*, now Fig 5D and have also added the *rnGAL4/+* control. We have also performed various hemi-control experiments as suggested by the reviewer to our best capabilities. We have added a separate graph with the hemicontrols in the as a Reviewer Response Figure 1.

Comment 16. Figure 7: Are IPCs necessary for the model? If not, I suggest removing them and placing the *Lgr3* neuron cell bodies much more anterior in this scheme. Their cell bodies are as anterior and rostral as it gets, approximately where the IPCs are depicted in this type of view of the CNS.

Response: We thank the reviewer for pointing this out and have removed IPCs from the figure, this figure is now labelled as Fig. 6.

Comment 17. Table S1- It would be preferable to see the data of these experiments, but if the authors prefer to show this data in a table, please at least add the dispersion analyses (eg standard deviation.. OR median+-quartiles OR Confidence intervals..), N of animals analysed, and statistics against controls.

Response: We thank the reviewer for pointing this out, we have added the number of larvae analysed, SEM values and statistics against the control condition.

Comment 18. In all figures with pupariation time: please also indicate significant findings in the graphs (with an asterisk, for instance) and adjust figure legends accordingly. This could facilitate understanding the data.

Response: Thanks for the suggestion. We have incorporated this information into figure legends.

Comment 19. Please revise Figure legends for punctuation.

Response: We have rectified all the errors in punctuation. We thank the reviewers for suggesting this.

Comment 20.

a) Abstract:

Line 10: What is the evidence to call *Dilp8* a "paracrine" factor?

Response: We thank the reviewer for pointing this out, we have changed the text to 'secreted factor'.

b) Introduction:

4th paragraph, 3rd sentence " Dilp8... buffers developmental noise and delays pupariation..." Buffering of developmental noise was first shown in Garelli et al., Science 2012, so this publication should be cited. 4th paragraph, 5th sentence: please include Jaszczak et al., Genetics 2016. This paper was published together with the 2015 papers, just a matter of timing that it got a 2016 date. Moreover, I do not think Katsuyama et al., 2015 is well cited to back up the statement in this sentence, hence I recommend removing that citation in this sentence.

Response: We thank the reviewer for pointing this out and have made necessary changes.

c) 6th paragraph: 5th line "targeting dilp8" : please specify if you mean the gene or the mRNA, or both. Same for line 7.

Response: We thank the reviewer for pointing this out and have made necessary changes.

d) Results Page 10, 1st paragraph, 1st sentence: the works cited are not the appropriate studies that demonstrated what is being stated. This was shown in Garelli et al., Science 2012 and Colombani et al., Science 2012. Results Page 10, 1st paragraph, line 11: Please also cite Colombani et al., Science 2012, who first showed that JNK is required for dilp8 regulation.

Response: We thank the reviewer for pointing this out and are extremely apologetic for this oversight. We have made necessary changes to the manuscript.

e) Discussion, 2nd paragraph, line 4: again, please indicate the rationale for using "paracrine" to describe Dilp8's activities. The current widely accepted model is that Dilp8 acts on interneurons in the brain (eg, reviewed in Juarez-Carreno et al., Cell Stress, 2018; Gontijo and Garelli, Mech Dev, 2018; Mirth and Shingleton, Front Cell Dev Biol, 2019; Texada et al., Genetics 2020; Boulan and Leopold, 2021). In order to reach the brain, Dilp8 has to be secreted from the discs and travel to the brain. This is as an endocrine mechanism as it gets for a small larva, considering that some discs can be on the opposite side of the larva (eg, genital discs). While this does not exclude that Dilp8 could also act paracrinally, the only evidence that I am aware of comes from other contexts such as during transdetermination (where Dilp8 has been proposed to work in an autocrine or paracrine fashion, via Drl in imaginal discs (Nemoto et al., Genes to Cells, 2023), however, this is not cited appropriately in this manuscript and is less related to the Lgr3-dependent pathway being studied here.

Response: We totally agree with the reviewer and appreciate clarifying this for us. We have made necessary changes to the text.

f) Discussion Page 13, 1st paragraph, This claim is supported by data presented in Garelli et al., Science 2012, not the other two papers. Garelli et al., 2015 shows that the Lgr3 receptor also participates in buffering developmental noise. Other studies have corroborated the Garelli et al., 2012 finding: eg, Colombani et al., Curr Biol 2015; Boone et al., Nat Commun 2016; Blanco-Obregon et al., Nat Commun 2022). Many other studies have shown that Dilp8 promotes developmental stability under tissue stress and challenges.

Discussion Page 12, 3rd paragraph, 2nd sentence: "The Lgr3 neurons directly interact with ... PTHH ...and insulin-producing neurons" Please cite Colombani et al., 2015 and Vallejo et al., Science 2015. Vallejo et al., propose that circuit with insulin-producing neurons. In the 3rd sentence, only Jaszczak et al., 2016 is cited, whereas this claim/ model comes from many studies, such as Halme et al., Curr Biol, 2010; Hackney et al., PLoS One 2012; Garelli et al. Science 2012; Colombani et al., Science, 2012; and the Lgr3 papers from 2015). Jaszczak et al., actually propose that Lgr3 is also required in the ring gland in addition to neurons.

Discussion page 14 last paragraph, 10 line, "In Aedes aegypti regulates ilp8 (Ling et al.,

2017)". As far as I understand mosquitoes do not have a dilp8 orthologue (see for instance Gontijo and Gontijo, Mech Dev 2018; and Jan Veenstra's work). ilp nomenclature (numbering) does not follow that of Drosophila, so ilp8 is probably a typical Insulin/IGF-like peptide and is NOT an orthologue of Dilp8, a relaxin, so this citation needs to be removed or placed into the broader context of microRNA regulation of ilps.

Response: We are really sorry for the numerous glaring errors in the references. We thank the reviewers for correcting this for us. We have made necessary changes to the text.

Thank you for the opportunity to review your interesting work, Alisson Gontijo and Rebeca Zanini
Reviewer #3 (Significance (Required)):

If the authors address these points raised above, we believe the manuscript should be a valuable contribution to the field, and help in the understanding of how tissues respond to growth aberrations and the regulation of transcript levels by microRNAs.

Author's concluding response:

We thank all the reviewers for the overall constructive comments and suggestions that we believe have helped us to improve our manuscript. We have incorporated all the changes suggested, especially regarding errors in citing key references. We have performed most of the experimental suggestions. Also, we have modified the way in which graphs are presented, including statistical tests as suggested by the reviewers. Several controls have been performed to strengthen the manuscript further. We believe that this review process aided in significantly improving this manuscript.

Reviewer #1

Evidence, reproducibility and clarity

SUMMARY

In this study, Fernandes and colleagues addressed the question of the role of micro-RNAs in regulating the coupling between organ growth and developmental timing. Using *Drosophila*, they identified the conserved micro-RNA miR-184 as a regulator of the developmental transition between juvenile larval stages and metamorphosis. This transition is under the control of the steroid hormone Ecdysone, and has been shown to be modulated in case of abnormal tissue growth to adjust the duration of larval growth in response to developmental perturbations. The relaxin-like hormone Dilp8 has been identified as a key secreted factor involved in this coupling. Here, the authors show that miR-184 is involved in the regulation of Dilp8 expression both in physiological conditions and upon growth perturbation. They propose that this function is carried out in imaginal tissues, where miR-184 levels are modulated by tissue stress. While several factors have already been involved in triggering sharp dilp8 induction at the transcriptional level, this study adds another level of complexity to the regulation of Dilp8 by proposing that its expression is fine-tuned post-transcriptionally through repression by miR-184.

MAJOR COMMENTS

Overall, the manuscript is well organized, and the logics of the experimental plan well presented. The results are clear, and I appreciate the quality of the pupariation curves. However, I believe that two main conclusions of the paper are not fully supported by the results presented in the figures: the direct regulation of dilp8 3'UTR by miR-184, and the specificity of this regulation in imaginal discs. Here I develop in more details these two aspects.

Comment 1) The strategy of the 3'UTR sensor is not fully optimized. Indeed, in most experiments, qRT-PCR is used to assess dilp8 expression levels, although it reflects both transcriptional and post-transcriptional. Importantly, to show that post-transcriptional regulation is involved in the response to tissue damage, the levels of the 3'UTR sensor should be analyzed in discs expressing RAcS (showing at the same time that the response is cell-autonomous in the discs). The expected upregulation of the sensor should be prevented by simultaneous expression of miR-184. This approach would shed light on the relative contribution of transcriptional versus post-transcriptional regulation of dilp8 in response to growth perturbation.

Response: We thank the reviewer for this comment. We agree that qRT-PCRs do not distinguish between transcriptional and post-transcriptional changes of *dilp8* levels, in response to changes in *miR-184* levels and tissue damage. In addition to the qRT-PCR data we have looked at *dilp8*-3'UTR-GFP reporter in response to overexpression of *miR-184* in the wingdisc using *patched*-Gal4 driver, which show downregulation of the GFP reporter in the *ptc* domain (Fig 4C-D'). This suggests that *dilp8* mRNA is a direct target of *miR-184* by post-transcriptional regulation through its 3'UTR. Further, to confirm the specificity of the effect of *miR-184* on *dilp8*-3'UTR, we generated a *dilp8*-3'UTR mutant in which the single target site for *miR-184* was mutated. We show that the mutated *dilp8*-3'UTR reporter doesn't show any regulation in response to *miR-184* overexpression in the *ptc* domain of the wingdisc (Fig. 4E, E', F, F'). This experiment confirms the specificity of the *dilp8*-3'UTR regulation by *miR-184*.

As suggested by the reviewer we analysed *dilp8*-3'UTR-GFP reporter expression by overexpressing RicinA using *ptcGAL4* driver in the wing imaginal disc (Fig. S6F-G'). We observed a slight but consistent increase in the *dilp8*-3'UTR-GFP reporter expression, indicating post-transcriptional regulation of *dilp8* expression in response to tissue damage. However, the increase of reporter GFP levels observed in this experiment in response to tissue damage is mild (Fig. S6F-G') than expected based on the qRT-PCR results (Fig S6A and B). We have added this new data to the manuscript (Fig. S6F-G').

We propose the following reasons to explain this result:

a) both transcriptional and post-transcriptional regulation of *dilp8* mRNA in response to developmental perturbations

b) the data on 3'UTR reporter GFP is specifically from the *ptc* domain expression of RicinA, whereas for *dilp8* transcript levels we have expressed RicinA in all larval imaginal tissues, or in the entire wing imaginal disc, which could be one of the reasons for the stronger effect seen on *dilp8* mRNA levels

c) we are not certain if the *tubulin*-promoter driven *dilp8*-3'UTR GFP reporter reflects post-transcriptional regulation of *dilp8* by miR-184 efficiently in comparison to qRT-PCR. This is especially as the reporter-GFP-3'UTR will be expressed at very high levels due to the *tubulin* promoter, a majority of this reporter-GFP mRNA may not be relieved from degradation due to the moderate suppression of miR-184 in response to RicinA overexpression.

Thus, our experiments suggest that *dilp8* levels are regulated post-transcriptionally by miR-184 which contributes to pupariation delays in response to tissue damage. In support of this, we could rescue pupariation delays and *dilp8* induction caused by RicinA expression using overexpression of *miR-184* (Figs 5B, C). Thus, we confirm that the effect of post-transcriptional regulation by *miR-184* during developmental perturbations also contributes to *dilp8* induction and pupariation delays. Unfortunately, due to experimental limitations we could not perform simultaneous expression of RicinA and *miR-184* to evaluate the rescue of *dilp8*-3'UTR-GFP sensor expression. The levels of *dilp8*-3'UTR sensor GFP is reduced efficiently by *miR-184* overexpression (Fig 4D), which prevented us from attempting the rescue of the moderate increase of *dilp8*-3'UTR GFP levels in response to RicinA.

Comment 2) In my opinion, the use of a 3'UTR sensor is not sufficient to conclude that the regulation by miR-184 is direct, as miR-184 could also regulate an intermediate factor that acts on dilp8 post-transcriptional regulation. To solve this issue, a common strategy is to generate a 3'UTR sensor with mutated binding sites that should abolish the regulation by miR-184. This mutated 3'UTR might also respond differently to tissue damage, which would strongly support the conclusions of the study.

Response: We couldn't agree more with the reviewer, this comment is addressed in the response to comment 1. We have confirmed the specificity of regulation of *dilp8*-3'UTR by miR-184 using target site mutated *dilp8*-3'UTR (new figures added to the manuscript Fig. 4E, E', F, F'). We tested if the changes in *dilp8* mRNA levels in response to tissue damage is post-transcriptional mediated by *miR-184*. We observe that there is a slight, but consistent increase of *dilp8*-3'UTR GFP reporter levels in the *ptc* domain of wingdisc in response to RicinA expression, suggesting a role for miR-184 mediated post-translational regulation of *dilp8*. However, we have not yet tested the mutated *dilp8*-3'UTR GFP reporter in response to tissue damage.

Comment 3) Concerning the tissue-specific regulation of Dilp8 by miR-184, these results need to be strengthened. Indeed, this comes mostly from phenotypes observed with *rn-GAL4*. Although this is a classical tool for driving expression in imaginal discs, *rn-GAL4* also drives strong expression in other tissues that could contribute to triggering a delay, such as the CNS and part of the gut (proventriculus). In our hands, some growth phenotypes in the wing obtained with *rn-GAL4* could be fully reverted by blocking GAL4 in the CNS indicating that the phenotype was not wing-specific. Importantly, miR-184 seems to be highly expressed in the CNS according to FlyBase, reinforcing the possibility that it plays a role in this organ. Here I propose approaches to confirm that miR-184 mediated regulation of *dilp8* and developmental timing indeed occur in the discs:

- Another driver with less secondary expression sites could be used (*pdmR11F02-GAL4*), or *rn-GAL4* could be combined with an *elav-GAL80* to prevent expression in most neurons.
- The authors could identify the source of Dilp8 upregulation in miR-184 mutants using tissue-specific qRT-PCR instead of whole larvae expression like in Fig 4A-B.
- This tissue-specific upregulation could be functionally tested using a rescue experiment, in which the delay observed in miR-184 mutants could be rescued by disc-specific downregulation of Dilp8 (using *pdm2-GAL4* for instance).

Response: We are thankful to the reviewer, and agree that it is important to show that the effects that we see using *rn-Gal4* are specific to imaginal discs, and not due to an effect in CNS. We tested

this by expressing miR-184 sponge in the CNS. Though miR-184 is highly expressed in the larval CNS, downregulation of miR-184 specifically in the pan-neuronal background using *elav*-GAL4 led to no effects on pupariation timepoint. We have added this as supplementary data Figure S4. Therefore, we believe that the *miR-184* downregulation phenotype in the *rnGAL4* background can be mainly attributed to its role in the imaginal discs. In addition, as suggested by the reviewer we have also demonstrated that downregulation of miR-184 in the imaginal discs using *rnGAL4* driver leads to an increase in *dilp8* expression (Fig S5B). Thus confirming that *dilp8* mRNA is enhanced in the imaginal discs by blocking *miR-184*.

OPTIONAL: Because it is known that *dilp8* is strongly regulated at the transcriptional level, the relative input from post-transcriptional upregulation is an important question arising from this study. Although it might be a more long-term approach, I believe that generating a *Dilp8* mutant lacking its 3'UTR or, even better, with mutated miR-184 binding sites, would shed light on the role of this regulation for the response to growth perturbation and/or developmental stability (fluctuating asymmetry).

Response: We thank the reviewer for the suggestion. This would have been an interesting experiment to carry out especially in the context of fluctuating asymmetry.

MINOR COMMENTS

1. I think that a number of results could be moved to SI as they are either controls, or reproduce published data without bringing novelty. For instance, results in Fig 5A-D are similar to data published by Sanchez et al, as stated in the text. Fig6A as well.

Response: We thank the reviewer for this suggestion, Fig. 5A-D, and F has been moved to Fig. S6A-E. We have also moved data from Fig. 6 to Fig. 5, as a result Fig 6 A-D has become Fig. 5 B-D.

2. Fig 6D is quite mysterious, as it suggests that basal JNK activation regulates miR-184, which is different from a context of tissue damage. I think that this result could be removed. Alternatively, if the authors want to dig in that direction, more experiments should be provided, such as *bskDN* expression in an RACs context and the effects on miR-184 levels and the 3'UTR sensor (since transcript levels are already published).

Response: We would like to clarify that our experiments suggest that endogenous JNK signalling negatively regulates *miR-184*, as blocking basal JNK signalling using *bskDN* increased the levels of *miR-184* (changed to Fig 5D). Enhanced JNK signalling has been reported to be involved in tissue damage responses, and we propose that RicinA mediated increase in JNK signalling leads to the reduction of *miR-184* (changed to Fig 5A, S6D-E). However, we are not strongly implying this as we did not co-express RicinA and *bskDN* to show that JNK signalling is responsible for the drop in miR-184 levels in response to tissue damage. We thank the reviewer for seeking this explanation, we have rewritten the results section to improve clarity.

3. The references related to *Dilp8* should be checked more in detail in the intro and discussion. About *Dilp8* and developmental stability: remove the ref to Colombani et al 2012, instead put Boone et al 2016 and add Blanco-Obregon et al 2022 (in addition to Garelli et al 2012 who initially identified this phenotype. About *Lgr3* as the receptor for *Dilp8*: add Colombani et al, Current Biology 2015, and cite here Vallejo et al 2015, Garelli et al 2015. Among the important transcriptional regulators of *Dilp8*, *Xrp1* could be mentioned (Boulan et al 2019, Destefanis et al 2022) as it plays a complementary function to JNK depending on the type of tissue stress.

Response: We are really sorry for the glaring errors in citing appropriate references. We thank the reviewer for correcting this for us. We have made necessary changes to the text.

Significance

GENERAL ASSESSMENT

This study provides convincing data showing that the conserved microRNA miR-184 plays a role in regulating developmental timing in *Drosophila* through modulating the levels of *Dilp8*, a key factor in the coupling between tissue growth and developmental transitions. The results are convincing,

but the general conclusions of the paper need to be strengthened regarding the direct regulation of *dilp8* by miR-184 and the tissue-specificity of this interaction.

ADVANCE

Dilp8 is a key factor that modulates growth and timing in response to developmental perturbations and contributes to developmental precision in physiological conditions. As such, its regulation has been studied by different groups in the last decade, leading to the identification of several inputs for its transcriptional regulation. Here, the authors uncover a post-transcriptional regulation by miR-184, adding another level of regulation of *Dilp8* that contribute to ensuring proper regulation of developmental timing, and opening the possibility that miR-184 might play similar roles in other species.

AUDIENCE

This study is of interest for researchers in the field of basic science, with a focus on developmental timing, tissue damage and biological function of microRNAs.

REVIEWER EXPERTISE

Drosophila, growth control, developmental timing, *Dilp8*.

Reviewer #2

Evidence, reproducibility and clarity

Drosophila has helped to characterize the mechanisms that coordinate tissue growth with developmental timing. The insulin/relaxin-like peptide *Dilp8* has been identified as a key factor that communicates the abnormal growth status of larval imaginal discs to neuroendocrine neurons responsible for regulating the timing of metamorphosis. *Dilp8*, derived from imaginal discs, targets four *Lgr3*-positive neurons in the central nervous system, activating cyclic-AMP signaling in an *Lgr3*-dependent manner. This signaling pathway reduces the production of the molting hormone, ecdysone, delaying the onset of metamorphosis. Simultaneously, the growth rates of healthy imaginal tissues slow down, enabling the development of proportionate individuals.

In this manuscript "miR-184 modulates *dilp8* to control developmental timing during normal growth conditions and in response to developmental perturbations" by Dr. Varghese and colleagues, the authors identify a new post transcriptional regulator of *Dilp8*. The authors show that miR-184 plays a pivotal role in tissue damage responses by inducing *dilp8* expression, which in turn delays pupariation to allow sufficient time for damage repair mechanisms to take effect.

Major points:

Comment 1) In most of the experiments for percentage of pupariation, the 50% pupariation in control is around 110 hours AED in figures 1, 2 and 3. In figures 5 and 6 using the UAS *Ricin*, the controls are more around 90 hours AED. Why this discrepancy?

Response: We thank the reviewer for asking for this clarification. The former experiments for Figs 1-3 were carried out at 25°C while the latter experiments with a cold sensitive version of *RicinA* (*UAS-RA^{CS}*), Figs 5 and 6 (now changed to Figs. 5 and S6 as suggested by reviewer #1) were carried out at 29°C (permissive temperature). This difference in temperature has led to alterations in pupariation timing. We apologise for not having mentioned this in the text, now we have made necessary corrections to the methods section clearly indicating this.

Comment 2) What is the mechanism behind the expression of miR-184 in stress conditions? Is miR-184 also implicated in other conditions giving rise to a developmental delay (X-rays irradiation or animal bearing *rasV12*, *scrib*^{-/-} tumors)?

Response: We thank the reviewer for these questions.

a) In response to developmental perturbations by *RicinA*, we believe that activation of JNK

signalling controls miR-184 expression. We propose this as our experiments show that imaginal disc damage leads to enhancement of JNK signalling and increase in *dilp8* mRNA levels (as reported earlier by Colombani et al 2012; Sánchez et al 2019), and a simultaneous reduction of *miR-184* (Figs. S6A, D, E). We also have performed new experiments to show that in response to RicinA expression in the wingdisc there is moderate increase in the *dilp8*-3'UTR-GFP sensor expression (Figs. S6F-G'), indicating a post-transcriptional regulation of *dilp8* expression in response to tissue stress. We also show that RicinA induced *dilp8* expression and pupariation delay can be rescued by increasing miR-184 levels (Fig 5B and C), suggesting that the reduction of miR-184 in response to tissue damage contributes to the damage responses. In a separate experiment we show that blocking the endogenous JNK pathway by the expression of *bskDN* enhances miR-184 levels, suggesting that miR-184 is under the regulation of JNK signalling (Fig 5D). Hence, we speculate that during tissue stress, activation of JNK signalling leads to a reduction of miR-184 levels which contributes to regulating the levels of *dilp8* post-transcriptionally and resulting in pupariation delays. The text has been modified to explain this better.

b) In a previous paper by Shu et al., 2017 (<https://doi.org/10.18632/oncotarget.22226>) decreased expression of miR-184 was observed in a *lglRNAi; RasV¹²* tumor background. Apart from this various studies have shown that *dilp8* levels increase in response to tumour, radiation stress, apoptosis, and tissue damage (Yeom et al 2021, Ray et al 2019, Demay et al 2014, Katsuyama et al 2015, Colombani et al 2012, Garelli et al 2012). Whether the regulation of *dilp8* by miR-184, occurs in these backgrounds is yet to be tested. We have now discussed this possibility in the manuscript.

Comment 3) *dilp8* mutant animals have also been shown to be more resistant to starvation or desiccation (<https://doi.org/10.3389/fendo.2020.00461>). Is miR-184 implicated in this answer?

Response: We thank the reviewer for this question. In our earlier experiments miR-184 has been demonstrated to be regulated by nutrition in the larval stages and lack of miR-184 led to enhanced larval death in response to diet restriction (Fernandes et al., 2022). miR-184 was also demonstrated to play a role in the insulin producing cells (IPCs) in regulating lifespan (Fernandes & Varghese., 2022). In the current work, we propose miR-184 to act upstream of *dilp8* in response to stress stimuli. Hence, it is possible that miR-184 might be involved in responses to starvation and desiccation stress in the adult female flies, by regulating *dilp8* levels post-transcriptionally. However, it has not been tested yet if the miR-184 regulation of *dilp8* plays a role in resistance to starvation or desiccation in adult females, as this was not within the scope of the current study. We have now added this reference in the discussion section.

Comment 4) *dilp8* expression has been also shown to be regulated by Xrp1 in response to ribosome stress (<https://doi.org/10.1016/j.devcel.2019.03.016>). This paper should be included in the manuscript. Is it possible that the expression levels of miR184 are regulated by Xrp1?

Response: We thank the reviewer for the suggestion and have incorporated the reference into the paper. During ribosome stress in the larval imaginal discs the stress-response transcription factor Xrp1 acts through *dilp8* in regulating systemic growth. We agree with the reviewer, it is possible that expression of miR-184 is regulated by Xrp1. Currently we have not explored this possibility. We have now added this to the discussion section.

Minor points:

1. Does the overexpression of miR184 induce an increased fluctuating asymmetry?

Response: We thank the reviewer for asking this question. The role of *dilp8* in the fluctuation asymmetry is only observed in the *dilp8* hypomorphic mutant background. To replicate this we would have to overexpress miR-184 in either the whole larvae or in the wing discs. Unfortunately overexpression of miR-184 in the wing discs (using *rnGAL4*) leads to pupal lethality while as overexpression of miR-184 in the whole larvae leads to embryonic lethality and therefore we were not be able to conclude from our experiments if miR-184 overexpression induces increased fluctuating asymmetry.

2. There are 2 references Colombani et al. (2012 for Dilp8 and 2015 for Lgr3). Can you double check that they are used accordingly

Response: We thank the reviewer for pointing these errors out and we have incorporated these changes into the paper.

Significance

Altogether, the paper present compiling lines of evidence supporting the proposed model. The experiments are well designed and are convincing. The papers is interesting and relevant for a broad audience.

Reviewer #3

Evidence, reproducibility and clarity (Required):

This is an interesting study demonstrating an interaction between miR-184 and the Drosophila insulin-like peptide 8 (*dilp8*) in the tissue damage response. The authors show that Dilp8 activity is negatively regulated by miR-184, apparently through direct interaction between miR-184 and the *dilp8*-3'UTR, which leads to lower *dilp8* mRNA transcript levels, via an undetermined mechanism, supposedly its degradation? Furthermore, the authors show that during aberrant tissue growth, miR-184 levels are very slightly downregulated (see comment below), and based on other experiments, imply causation of this with the increased *dilp8* mRNA levels that occur in these tissues, again via an unclear mechanism: upregulation or stabilization of *dilp8* mRNA. The authors present evidence that the JNK pathway, which had been known to be critical for *dilp8* mRNA upregulation upon tissue damage, does so via miR-184.

Major Comments:

Comment 1: The data showing the direct regulation of *dilp8*-3'UTR by miR-184 are not very strong and would require more controls to strengthen the claim, as described below.

Response: We have performed new experiments to validate that *dilp8*-3'UTR is regulated by miR-184. Please see the detailed responses to comments 10-12 below.

Comment 2: The miR-184 effects are also very small (less than 2-fold reduction with tissue damage; or less than 2-fold induction with JNK-pathway inhibition via *bskDN*). These two points are the weakest part of the manuscript and model.

Response: We agree with the reviewers on this point. The reduction in miR-184 levels in response to RicinA^{CS} expression is modest (25-30%), and the induction of miR-184 in response to *bskDN* expression is less than two-fold (Figs. 5A and D). In contrast, *dilp8* transcript levels increase several-fold in response to RicinA^{CS} expression (Fig. 5C, S6A and B). Since we measure *dilp8* transcript levels by qPCR, we detect both transcriptional and post-transcriptional contributions to *dilp8* regulation. In addition, we have performed a new experiment to check the post-transcriptional regulation of *dilp8*, in response to tissue damage. Though the change in the *dilp8*-3'UTR GFP reporter upon RicinA^{CS} expression in the *ptc* domain of the wingdisc is mild (Figs. S6F-G'), this strongly suggests a post-transcriptional outcome of the reduction of miR-184 levels on *dilp8*. Hence, we propose that tissue damage induces strong transcriptional activation of *dilp8*, while the reduction of miR-184, despite its smaller magnitude, contributes to *dilp8* upregulation via post-transcriptional regulation. In support of this, our experiments demonstrate direct regulation of the *dilp8*-3'UTR by miR-184 (Figs. 4C-F'), and show strong *dilp8* mRNA upregulation in miR-184 deficient conditions (Fig. 4A and B), suggesting the role of miR-184 in maintaining *dilp8* levels. We also show that RicinA^{CS} induced effects on *dilp8* and pupariation delay are reversed by co-expression of miR-184 (Fig. 5C). We do not claim that regulation by miR-184 is the sole mechanism for driving *dilp8* induction during tissue damage, but suggest that miR-184-mediated post-transcriptional regulation acts in a complementary manner to transcriptional responses. Furthermore, we believe that the mild effect of JNK signaling on miR-184 (as shown by the *bskDN* experiment) is sufficient for the moderate reduction of miR-184 in response to tissue damage.

Comment 3: Regarding the expression levels, it does not help that the authors show bar graphs with standard errors of the mean instead of the actual data points to allow reliable appreciation of the data dispersion.

Response: We have modified our figures and have performed statistical analysis according to the suggestions of the reviewers, please see responses to comments 1-9, and 13-19.

Comment 4: It is difficult to understand how minute changes in miR-184 levels can lead to over an order of magnitude differences (in some cases) in *dilp8* mRNA levels considering that it is a stoichiometric relationship. Maybe ?miR-184-Dicer1? complexes are highly stable and re-used for multiple *dilp8* transcripts - the authors could discuss how they understand this occurring in their manuscript.

On the same line, discussion is also rather weak on what regards the mechanism of control of *dilp8* mRNA levels by miR-184. Please discuss eg, the evidence for mRNA degradation induction by microRNAs with this UTR binding profile (imperfect UTR binding Fig S4) and-if appropriate-how other possible regulatory models (direct and indirect) could explain the findings.

Response: We accept the reviewers comment that 25-30% reduction of miR-184 is low in comparison to the many fold increase in *dilp8* levels. We believe that both post-transcriptional and transcriptional changes are responsible for the induction of *dilp8* in response to tissue damage. However, our experiments suggest the role of post-transcriptional regulation by miR-184, as pupariation delay is rescued by miR-184 overexpression (also please see the response to the previous comment). We are not ruling out the possibility of transcriptional regulation of *dilp8* mRNA, rather we are suggesting the possibility that both transcriptional and post-transcriptional means are responsible for changes in *dilp8*. Moreover, we have not performed absolute measurement of miR-184 in the imaginal discs (what we show is a comparison between control and RicinA^{CS} expression), hence we do not have an exact estimate of how many miR-184 molecules are reduced and if they would be greatly equal or more in comparison to the *dilp8* mRNA molecules that are upregulated, as again while measuring *dilp8* mRNA we are not checking how many molecules of *dilp8* exactly are increased. As the reviewer suggests, it is possible that miR-184-RISC could be stable to handle multiple *dilp8* molecules one after the other, hence it is not a 1:1 relationship between miR-184:*dilp8*. We have included this in the manuscript. It is also known that imperfect 3'UTR binding as seen in most animal microRNAs leads to translational repression and mRNA deadenylation, which eventually results in mRNA degradation.

Comment 5: We suggest the authors carefully revise their citations to cite appropriate work that supports the claims, and also to avoid missing the seminal studies that report the claims they cite.

Response: We are really apologetic for the errors citing the key references. We are grateful to the reviewers for correcting this for us. We have made changes to the text to include and correct the references.

We have the suggestions below which we hope will help the authors improve their manuscript. If the authors address these points raised above, we believe the manuscript should be a valuable contribution to the field, and help in the understanding of how tissues respond to growth aberrations and the regulation of transcript levels by microRNAs.

Detailed Comments:

Comment 1. Results 1st paragraph: please describe the screen in more detail. As written, one only discovers it was a miRNA loss-of-function screen when reading the legend of Table S1. Please show the original data of the screen - with dispersion if possible.

Response: We thank the reviewers for these suggestions, we have now included the data from the screen with SEM, and p-values.

Comment 2. Results 1st paragraph, Fourth line, "While several miRNAs caused delays in pupariation by 12 hours or more..". Please correct, as actually loss of miRNAs caused delays.

Response: We thank the reviewer for pointing out this error, we have corrected the text

accordingly.

Comment 3. Results (Figure 1) - It says that data from three independent experiments are shown. However there is no dispersion in the data. Could the authors please explain this? Are the results of the three experiments summed and presented as one? or is this one of the three?

Response: We thank the reviewers for these suggestions and have plotted data with the SEM values.

Comment 4. It is reported in the legend of Figure S2 that LogRank test was performed to determine statistical significance. However, no statistical data is presented. Please show the results.

Response: We thank the reviewers for these suggestions to improve the data presentation, we have incorporated the p-value as suggested.

Comment 5. Fig2A and B. Please show the data points in the bar graphs (as in Figure. 2C), or choose another data representation. Please consider redoing statistical analysis with a simple t-test. It is not clear to me why ANOVA was used to compare two samples. Please state that data are normalized also to control (tub-GAL4>UAS-scramble). Please state the h post-hatching from which the RNA samples were collected (as in Fig 2C for 20HE quantification).

Response: We thank the reviewers for these suggestions to improve the data presentation, we have incorporated all changes as suggested. Similar changes have been incorporated to the rest of the figures of the manuscript as well. Hours post-hatching information for each figure is now added to the figure legends.

Comment 6. Fig2C. Fig legend states the bar graphs are "absolute values". Please specify if the bar represents the average, median or something else.

Response: We thank the reviewer for pointing this out, we have made the suggested changes.

Comment 7. Throughout the manuscript: please use GAL4 in capital letters or at least standardize it throughout the ms. Currently there are GAL4s and Gal4s.. eg compare Fig 2 and 3 legends.

Response: We thank the reviewer for pointing this out, we have incorporated all changes as recommended.

Comment 8. FigS3A and B. Please revise as Fig2A and B above. and apply the same criteria in the respective figure legend.

Response: We thank the reviewer for pointing this out, we have made the changes as recommended.

Comment 9. Fig. 4 - please indicate on the figures what is whole larvae and what is wing imaginal discs. This will facilitate understanding of the figure.

Response: We thank the reviewers for these suggestions and have included this information in all the figures.

Comment 10. Fig 4 - Data - Authors do not show that rn-GAL4>miR-184-sponge causes up regulation of dilp8 mRNA levels, hence the model is weakened. Doing this experiment would significantly strengthen the study whatever the result is.

Response: We thank the reviewer for pointing this out and we have included this in the manuscript (Fig S5B).

Comment 11. The *dilp8*-3'UTR experiment is weak especially because its generation is not sufficiently well described in the manuscript. "The *dilp8* 3'UTR-GFP reporter line was created as described in (Vargheese & Cohen, 2007)" is not sufficient. Please describe the construct generation in sufficient detail so that the experiments can be reproduced by others.

Response: We thank the reviewer for pointing this out and we have elaborated in the methods section on how we generated the *dilp8* 3'UTR-GFP reporter and *dilp8* 3'UTR mutant GFP reporter lines. The plasmid was originally created in Steve Cohen's lab at EMBL, by modifying pCasper4 plasmid, by introducing a *tubulin* promoter, EGFP and a multiple cloning site, which allows one to clone 3'UTRs of target genes into this plasmid. *Not1* and *Xho1* sites were used to clone the *dilp8*-3'UTR and mut-3'UTR. We hope this explains our strategy sufficiently.

Comment 12. Making assumptions, if the construct is as described in Vargheese & Cohen, 2007 and contains all of the *dilp8* 3'UTR - it should be a Tubulin-driven GFP gene with a *dilp8*-3'UTR "Tub-GFP-(*dilp8* 3'UTR)". In this case the authors need to rule out the alternative interpretation of the result in Fig. 4D by showing that the expression of miR-184 does not down regulate Tub-GFP expression itself. The best scenario would be to have a mutated *dilp8* 3'UTR for the miR-184 recognition site. This experiment would significantly strengthen the study and model.

Response: We thank the reviewer for pointing this out. We agree with the reviewers that this experiment is needed to prove direct regulation of the *dilp8*-3'UTR by miR-184. We have mutated the sequences complementary to the seed region of miR-184 in the *dilp8*-3'UTR, and demonstrated that overexpression of miR-184 does not regulate the mutated *tub*-GFP-(*dilp8* 3'UTR) expression. This confirms that the *dilp8* gene is a direct target of *miR-184*. This data is added to the manuscript as Figs 4E-F'.

Comment 13. Figure 4C-D please separate *dilp8* from 3'UTR with a space or hyphen.

Response: We thank the reviewer for pointing this out and have separated *dilp8* from 3'UTR with a hyphen.

Comment 14. Figure 4E. Please name the *dilp8* allele as M100727 as it is not a KO, but rather a hypomorphic mutation (fully WT *dilp8* transcripts are still generated, albeit at a much lower level).

Response: We thank the reviewer for pointing this out and we have made the necessary changes.

Comment 15. Figure 6D: please add UAS to *bskDN/+*. All figures have *rn*-GAL4 alone or with UAS-GFP as control. This finding would be strengthened with this other control, especially because the size effect is small. This being said a general comment for all experiments is that hemi-controls are generally missing for all figures. eg, in Fig 3. One would typically include controls such as A. *Phm*>+ and +>miR.184; B. *aug21*>+ and +>miR.184; C. *ptth*>+ and +>miR.184; D. *rn*>+ and +>miR.184

Response: We thank the reviewer for pointing this out. We have added UAS to *bskDN*, now Fig 5D and have also added the *rn*GAL4/+ control. We have also performed various hemi-control experiments as suggested by the reviewer to our best capabilities. We have added a separate graph with the hemicontrols in the as a Reviewer Response Figure 1.

Comment 16. Figure 7: Are IPCs necessary for the model? If not, I suggest removing them and placing the *Lgr3* neuron cell bodies much more anterior in this scheme. Their cell bodies are as anterior and rostral as it gets, approximately where the IPCs are depicted in this type of view of the CNS.

Response: We thank the reviewer for pointing this out and have removed IPCs from the figure, this figure is now labelled as Fig. 6.

Comment 17. Table S1- It would be preferable to see the data of these experiments, but if the authors prefer to show this data in a table, please at least add the dispersion analyses (eg standard deviation.. OR median+-quartiles OR Confidence intervals..), N of animals analysed, and statistics against controls.

Response: We thank the reviewer for pointing this out, we have added the number of larvae analysed, SEM values and statistics against the control condition.

Comment 18. In all figures with pupariation time: please also indicate significant findings in the graphs (with an asterisk, for instance) and adjust figure legends accordingly. This could facilitate understanding the data.

Response: Thanks for the suggestion. We have incorporated this information into figure legends.

Comment 19. Please revise Figure legends for punctuation.

Response: We have rectified all the errors in punctuation. We thank the reviewers for suggesting this.

Comment 20.

a) Abstract:

Line 10: What is the evidence to call Dilp8 a "paracrine" factor?

Response: We thank the reviewer for pointing this out, we have changed the text to 'secreted factor'.

b) Introduction:

4th paragraph, 3rd sentence " Dilp8... buffers developmental noise and delays pupariation..." Buffering of developmental noise was first shown in Garelli et al., Science 2012, so this publication should be cited. 4th paragraph, 5th sentence: please include Jaszczak et al., Genetics 2016. This paper was published together with the 2015 papers, just a matter of timing that it got a 2016 date. Moreover, I do not think Katsuyama et al., 2015 is well cited to back up the statement in this sentence, hence I recommend removing that citation in this sentence.

Response: We thank the reviewer for pointing this out and have made necessary changes.

c) 6th paragraph: 5th line "targeting dilp8" : please specify if you mean the gene or the mRNA, or both. Same for line 7.

Response: We thank the reviewer for pointing this out and have made necessary changes.

d) Results Page 10, 1st paragraph, 1st sentence: the works cited are not the appropriate studies that demonstrated what is being stated. This was shown in Garelli et al., Science 2012 and Colombani et al., Science 2012. Results Page 10, 1st paragraph, line 11: Please also cite Colombani et al., Science 2012, who first showed that JNK is required for dilp8 regulation.

Response: We thank the reviewer for pointing this out and are extremely apologetic for this oversight. We have made necessary changes to the manuscript.

e) Discussion, 2nd paragraph, line 4: again, please indicate the rationale for using "paracrine" to describe Dilp8's activities. The current widely accepted model is that Dilp8 acts on interneurons in the brain (eg, reviewed in Juarez-Carreño et al., Cell Stress, 2018; Gontijo and Garelli, Mech Dev, 2018; Mirth and Shingleton, Front Cell Dev Biol, 2019; Texada et al., Genetics 2020; Boulan and Leopold, 2021). In order to reach the brain, Dilp8 has to be secreted from the discs and travel to the brain. This is as an endocrine mechanism as it gets for a small larva, considering that some discs can be on the opposite side of the larva (eg, genital discs). While this does not exclude that Dilp8 could also act paracrinally, the only evidence that I am aware of comes from other contexts such as during transdetermination (where Dilp8 has been proposed to work in an autocrine or paracrine fashion, via Drl in imaginal discs (Nemoto

et al., *Genes to Cells*, 2023), however, this is not cited appropriately in this manuscript and is less related to the Lgr3-dependent pathway being studied here.

Response: We totally agree with the reviewer and appreciate clarifying this for us. We have made necessary changes to the text.

f) Discussion Page 13, 1st paragraph, This claim is supported by data presented in Garelli et al., *Science* 2012, not the other two papers. Garelli et al., 2015 shows that the Lgr3 receptor also participates in buffering developmental noise. Other studies have corroborated the Garelli et al., 2012 finding: eg, Colombani et al., *Curr Biol* 2015; Boone et al., *Nat Commun* 2016; Blanco-Obregon et al., *Nat Commun* 2022). Many other studies have shown that Dilp8 promotes developmental stability under tissue stress and challenges.

Discussion Page 12, 3rd paragraph, 2nd sentence: "The Lgr3 neurons directly interact with ... PTHH ...and insulin-producing neurons" Please cite Colombani et al., 2015 and Vallejo et al., *Science* 2015. Vallejo et al., propose that circuit with insulin-producing neurons. In the 3rd sentence, only Jaszczak et al., 2016 is cited, whereas this claim/ model comes from many studies, such as Halme et al., *Curr Biol*, 2010; Hackney et al., *PLoS One* 2012; Garelli et al. *Science* 2012; Colombani et al., *Science*, 2012; and the Lgr3 papers from 2015). Jaszczak et al., actually propose that Lgr3 is also required in the ring gland in addition to neurons.

Discussion page 14 last paragraph, 10 line, "In *Aedes aegypti* regulates ilp8 (Ling et al., 2017)". As far as I understand mosquitoes do not have a dilp8 orthologue (see for instance Gontijo and Gontijo, *Mech Dev* 2018; and Jan Veenstra's work). ilp nomenclature (numbering) does not follow that of *Drosophila*, so ilp8 is probably a typical Insulin/IGF-like peptide and is NOT an orthologue of Dilp8, a relaxin, so this citation needs to be removed or placed into the broader context of microRNA regulation of ilps.

Response: We are really sorry for the numerous glaring errors in the references. We thank the reviewers for correcting this for us. We have made necessary changes to the text.

Thank you for the opportunity to review your interesting work,

Alisson Gontijo and Rebeca Zanini

Reviewer #3 (Significance (Required)):

If the authors address these points raised above, we believe the manuscript should be a valuable contribution to the field, and help in the understanding of how tissues respond to growth aberrations and the regulation of transcript levels by microRNAs.

Author's concluding response:

We thank all the reviewers for the overall positive comments and suggestions that we believe have helped us to improve our manuscript. We have incorporated all the changes suggested, especially regarding errors in citing key references. We have performed most of the experimental suggestions. Also, we have modified the way in which graphs are presented, including statistical tests as suggested by the reviewers. Several controls have been performed to strengthen the manuscript further. We believe that this review process aided in significantly improving this manuscript.

Original submission

First decision letter

MS ID#: dev.205280

MS Title: miR-184 modulates dilp8 to control developmental timing during normal growth conditions and in response to developmental perturbations

Authors: Jervis Fernandes, Muhammed Naseem, Ayisha Marwa and Jishy Varghese

Dear Dr Varghese,

Thank you for sending your manuscript to Development through Review Commons.

I have now received all the referees reports on the above manuscript, and have reached a decision. The referees' comments are appended below.

The overall evaluation is very positive and we would like to publish your manuscript in Development. While two of the reviewers accept your manuscript following revision, reviewer 2 makes several comments. I looked at the comments carefully and especially their comment 1 about your new control experiment to check the direct regulation of dilp8. I would argue that given that you have an internal control for fluorescence in the Patched domain, it is not necessary to compare multiple insertions or to check that all constructs are in the same insertion site. All in all I believe the control is great and is sufficient to support your conclusions. Please address the few other points of clarification from Reviewer 2. If you do not agree with any of their criticisms or suggestions explain clearly why this is so. Please send us a point-by-point response indicating your answers to the questions/comments from reviewer 2. I will assess your response myself.

Reviewer 1

Drosophila has helped to characterize the mechanisms that coordinate tissue growth with developmental timing. The insulin/relaxin-like peptide Dilp8 has been identified as a key factor that communicates the abnormal growth status of larval imaginal discs to neuroendocrine neurons responsible for regulating the timing of metamorphosis. Dilp8, derived from imaginal discs, targets four Lgr3-positive neurons in the central nervous system, activating cyclic-AMP signaling in an Lgr3-dependent manner. This signaling pathway reduces the production of the molting hormone, ecdysone, delaying the onset of metamorphosis. Simultaneously, the growth rates of healthy imaginal tissues slow down, enabling the development of proportionate individuals.

In this manuscript "miR-184 modulates dilp8 to control developmental timing during normal growth conditions and in response to developmental perturbations" by Dr. Varghese and colleagues, the authors identify a new post transcriptional regulator of Dilp8. The authors show that miR-184 plays a pivotal role in tissue damage responses by inducing dilp8 expression, which in turn delays pupariation to allow sufficient time for damage repair mechanisms to take effect.

Altogether, the paper present compiling lines of evidence supporting the proposed model. The experiments are well designed and convincing. The papers is interesting and relevant for a broad audience. The authors have addressed most of my comments and the comments of the 2 others reviewers in Review Commons. I think that the paper is fulfilling the requirements for publication.

Minor point:

There is still problem in the listing of the references, Boulan et al., 2019 is missing in the references whereas Agarwal et al., 2018 is referenced but not existing in the text.

Reviewer 2

Fernandes et al. present a convincing study showing that in the absence of miR-184 in *Drosophila melanogaster* larvae, an imaginal-disc tissue stress response is triggered that delays pupariation via upregulation of *dilp8*. The weakest part of the study is when they attempt to demonstrate that miR-184 regulates *Dilp8* activity directly, via direct interaction with the 3'UTR of *dilp8*. The strength of these experiments is compromised by the UTR reporter tools used, which could be subject to local insertional artifacts, namely different insertions can be influenced by different cis-regulatory elements. The authors should either address this issue experimentally or avoid claims of direct regulation. Apart from this, there are important issues with figures and data analyses that should be clarified.

Major comments

Tub-GFP-*dilp8*-3'UTR constructs. Please confirm that these constructs were generated by random P element insertion, and if so, please add this to the methods section. Please also include at least the chromosome of the insertions used. The major problem with this random insertion is that the different insertions can suffer the influence of different cis regulatory elements nearby the different insertion sites. E.g., the Tub-GFP-*dilp8*-mut-3'UTR can be regulated by different elements than the Tub-GFP-*dilp8*-3'UTR. Can the authors exclude this? One way the authors could overcome this, is by inserting the constructs at the same attP site. Alternatively, if the authors could show that independent insertions of each construct behave similarly, this would strengthen their model. If the authors cannot do any of this, they should acknowledge this important imitation of their work, conclusions, and discussion, down-toning the conclusions of direct regulation of *dilp8* transcript levels by miR-184.

Fig. EV6 - the Tub-GFP-*dilp8*-3'UTR construct seems to be induced by UAS-RA[cs] in the patched-GAL4 strip, but also in other regions stained by anti-*ptc*, so likely patched-GAL4 is also activated there. Why would the Tub-GFP-*dilp8*-3'UTR respond positively to UAS-RA[cs]? The authors claim that this is because Ricin expression causes a down regulation of miR-184, hence the Tub-GFP-*dilp8*-3'UTR construct is unregulated. However, can the results also be explained by *dilp8*-3'UTR containing an enhancer for *dilp8* or alternatively, and more concerning, that Tub-GFP-*dilp8*-3'UTR is inserted at a locus that responds similarly to *dilp8* upon Ricin and miR-184 expression. If this is true, the Tub-GFP-*dilp8*-mut-3'UTR shouldn't respond to Ricin, but this result is of limited value since it is supposedly inserted in another random site and could suffer the influence of other cis-regulatory elements.

Table S1. This table is confusing. In the Table S1 legend it says that the mean pupariation time is plotted, but no mean is shown. Rather a median is presented. Please clarify. If it is the median, please clarify what the median pupariation time represents. Is it the median of all larvae of pulled experiments or the median of medians or average of medians from N experiments? This is important as these values are not necessarily the same. Based on which values was the SEM calculated? SEM is related to an average value, not the median. Can you please clarify? It would be much preferable to include another dispersion value of the median pupariation time, if these are for pulled larvae from different experiments (e.g., 25-75th percentiles). Or show the average of the medians \pm SEM in addition to the median. As it is, the table is not completely useful. No need to show so many decimal cases. It would be fine to round it to 1 decimal case.

Figure Legend 1: there is conflicting info in A and B, in A it says that there are four independent repeats. After it says that "Data from five independent experiments are shown for A and B; ($p < 0.0001$)."
Please clarify.

Figure 2A and B. Please check the error bars. They are not symmetric, which shouldn't happen in a linear Y axis.

Figure 2A - phm. Please verify the values and calculations. It is impossible that the average of the displayed three dots correspond to an average of 1, so something is wrong either with the

normalization, the data, the bar, or the scale? I suspect the same problem occurs in the all other values - eg, for dib, there is no way the average is 1 there (values are $>0.5 + >0.5 + >2.0$, which will surely give >1 as average). Please revise the calculations and normalizations used for plotting the graphs (and verify if they were used for statistics). This seems like a systemic problem present in all panels reporting results from qPCR experiments.

Figs. 4A,B; 5A,C,D; EV1; EV3; EV5B; EV6A,B,D & E. Again please verify error bars (they are not symmetrical) and the normalization/averaging of the data. Something is wrong either with the data, the bar, or the scale. For reference, please compare to Fig. 2C (Ecdysone measurements in larvae), which shows reasonable dispersion of the values around the mean value of the bar, with accordingly symmetrical error bars.

Please add the controls (GAL4 and UAS alone) presented in the point-by-point response to the respective graphs in the manuscript, with the adjusted corresponding statistical analyses.

Minor comments

Results, Page 8, 2nd paragraph "The imaginal discs have been shown to regulate pupariation timing, especially in response to growth perturbations.." Please include seminal references that show this. eg, work by Pat Simpson (Simpson et al 1980), and Alex Shingleton's lab (eg, Stieper et al., 2008).

Methods:

Pupariation assays: please indicate the number of repeats here as well apart from the legends. Please indicate the real range of animals used. It says 50, but Fig1A for instance says 142 and 104 animals for two genotypes with 4 independent repeats, so they cannot have been repeats of 50 animals.

Same for Fig 1B, which indicates 955 and 617 animals for "five independent experiments", which means they had much more than 50 animals each.

Methods,

Fly stocks: something is grammatically missing in the last sentence of this paragraph.

Methods,

Different symbols for degrees are used. Please standardize.

Methods,

Sample collection for RNA isolation. Here it says that larvae were collected 116 hrs post-hatching, but in the Figure 2 it says 112 hrs. Please revise.

Methods,

qRT-PCR: here it says that "The level of the genes was normalized to the level of actin." However, in the Figure 2 (and other qRT-PCR figures), for instance, it says that they were normalized to rp49. No actin primers are described. Please revise.

Please explain data analyses and plotting more clearly here.

Methods, microRNA qRT-PCR: please briefly explain the procedure and conditions used, including normalization. (eg, 2SrRNA is not mentioned here). The methods should stand alone and this is important for reproducibility, as kit instructions might be or become unavailable for others.

Thank you for the opportunity to revise your work.

With kind regards,

Alisson Gontijo and Rebeca Zanini

Reviewer 3

The authors have thoroughly addressed all my comments. In particular, the new results concerning the 3'UTR sensor (and its mutated version) significantly improve the manuscript. I therefore recommend its publication in Development.

Author response to reviewers' comments

Responses to Review Comments:

Reviewer 1:

Drosophila has helped to characterize the mechanisms that coordinate tissue growth with developmental timing. The insulin/relaxin-like peptide Dilp8 has been identified as a key factor that communicates the abnormal growth status of larval imaginal discs to neuroendocrine neurons responsible for regulating the timing of metamorphosis. Dilp8, derived from imaginal discs, targets four Lgr3-positive neurons in the central nervous system, activating cyclic-AMP signaling in an Lgr3-dependent manner. This signaling pathway reduces the production of the molting hormone, ecdysone, delaying the onset of metamorphosis. Simultaneously, the growth rates of healthy imaginal tissues slow down, enabling the development of proportionate individuals.

In this manuscript "miR-184 modulates dilp8 to control developmental timing during normal growth conditions and in response to developmental perturbations" by Dr. Varghese and colleagues, the authors identify a new post transcriptional regulator of Dilp8. The authors show that miR-184 plays a pivotal role in tissue damage responses by inducing dilp8 expression, which in turn delays pupariation to allow sufficient time for damage repair mechanisms to take effect.

Altogether, the paper present compelling lines of evidence supporting the proposed model. The experiments are well designed and convincing. The paper is interesting and relevant for a broad audience. The authors have addressed most of my comments and the comments of the 2 other reviewers in Review Commons. I think that the paper is fulfilling the requirements for publication.

1. Minor point:

There is still problem in the listing of the references, Boulan et al., 2019 is missing in the references whereas Agarwal et al., 2018 is referenced but not existing in the text.

Response: We thank the reviewer for pointing out these errors, we have made necessary changes to the manuscript. We are deeply grateful for your recommendation that the paper fulfils the requirement for publication.

Reviewer 2:

Fernandes et al. present a convincing study showing that in the absence of miR-184 in *Drosophila melanogaster* larvae, an imaginal-disc tissue stress response is triggered that delays pupariation via upregulation of dilp8. The weakest part of the study is when they attempt to demonstrate that miR-184 regulates Dilp8 activity directly, via direct interaction with the 3'UTR of dilp8. The strength of these experiments is compromised by the UTR reporter tools used, which could be subject to local insertional artifacts, namely different insertions can be influenced by different cis-regulatory elements. The authors should either address this issue experimentally or avoid claims of direct regulation. Apart from this, there important issues with figures and data analyses that should be clarified.

Response: We thank the reviewer for raising this point. A detailed explanation is given as a response to the major comments 1.

Major comments

Comment 1. Tub-GFP-dilp8-3'UTR constructs. Please confirm that these constructs were

generated by random P element insertion, and if so, please add this to the methods section. Please also include at least the chromosome of the insertions used. The major problem with this random insertion is that the different insertions can suffer the influence of different cis regulatory elements nearby the different insertion sites. E.g., the Tub-GFP-dilp8-mut-3'UTR can be regulated by different elements than the Tub-GFP-dilp8-3'UTR. Can the authors exclude this? One way the authors could overcome this, is by inserting the constructs at the same attP site. Alternatively, if the authors could show that independent insertions of each construct behave similarly, this would strengthen their model. If the authors cannot do any of this, they should acknowledge this important imitation of their work, conclusions, and discussion, down-toning the conclusions of direct regulation of dilp8 transcript levels by miR-184.

Response: We thank the reviewer for raising this point and we have added the details of the transgenic lines in the methods section. We agree that different insertions can be influenced by different cis-regulatory elements. However, we believe that our experiments demonstrate a direct regulation of dilp8 by miR-184 through its 3'UTR. We would like to present the following facts to justify this:

a) The endogenous expression pattern of the wildtype and mutant dilp8-3'UTR GFP in the larval wingdisc is not very different from each other, suggesting that both lines are subject to similar gene expression regulation and are not affected differently by their insertional background. If insertional site was preventing miR-184 from acting on the mut-3'UTR (leading to a lack of regulation), then there should have been a change in the levels of the reporter GFP expression owing to non-regulation by endogenous miR-184 expressed in the wing disc, which is not observed. Upon overexpression of miR-184 in the patched-domain we see a reduction of the GFP expression of the wildtype 3'UTR, specifically in this domain. A similar overexpression of miR-184 does not affect the mutant dilp8-3'UTR. Thus, the GFP expression from the control as well as mutant 3'UTR reporter lines does not vary drastically except in the ptc-domain upon overexpression of miR-184, despite the possible insertional site difference. This strongly suggests that an effect of the insertional site if any is minimal on the dilp8-3'UTR.

b) Both enhancement of miR-184 by its overexpression and suppression of miR-184 levels by expression of Ricin^{CS} in the wingdisc, shows opposing effects at the level of GFP expression in the wildtype-dilp8-3'UTR reporter, indicating a direct regulation of dilp8 by the microRNA. However, both treatments do not affect the mutant-3'UTR reporter GFP levels (FigS6 and Response Fig). This strongly suggests that insertional site do not play a role in the current 3'UTR reporter assay.

c) In this experiment we also have an internal control for fluorescence in the Patched domain (as we express our transgenes of interest only in the patched domain of the wing disc and not in the entire disc. Therefore we believe the control here sufficiently supports our conclusions.

d) We have shown several experiments to prove the regulation of dilp8 mRNA levels and resultant pupariation phenotypes by manipulating miR-184 levels during normal development and developmental damage induced effects. Thus, while we accept that there is a possibility that the insertional site of the dilp8-mutant-3'UTR reporter can deter regulation by miR-184, all the data that we provide strongly supports the regulation of dilp8 by miR-184.

Thus we propose that our reporter assays confirm a direct regulation of dilp8 through its 3'UTR by miR-184.

Kindly note that in the current scenario it is not feasible for us to redo the experiment within any reasonable timeline as the constructs will have to be introduced into a new attB plasmid vector and new transgenic flies need to be generated. We do not have an easy way to do this due to lack of microinjection facility at our institute, and the facility provided at another institute has a long queue, hence we are unable to perform the experiment using the same attP site as suggested by the reviewer.

Comment 2. Fig. EV6 - the Tub-GFP-dilp8-3'UTR construct seems to be induced by UAS-RA[cs] in

the patched-GAL4 strip, but also in other regions stained by anti-ptc, so likely patched-GAL4 is also activated there. Why would the Tub-GFP-dilp8-3'UTR respond positively to UAS-RA[cs]? The authors claim that this is because Ricin expression causes a down regulation of miR-184, hence the Tub-GFP-dilp8-3'UTR construct is unregulated. However, can the results also be explained by dilp8-3'UTR containing an enhancer for dilp8 or alternatively, and more concerning, that Tub-GFP-dilp8-3'UTR is inserted at a locus that responds similarly to dilp8 upon Ricin and miR-184 expression. If this is true, the Tub-GFP-dilp8-mut-3'UTR shouldn't respond to Ricin, but this result is of limited value since it is supposedly inserted in another random site and could suffer the influence of other cis-regulatory elements.

Response: *We thank the reviewers for this comment. Please see the detailed response to comment 1. We have performed an experiment to check the dilp8-mut-3'UTR reporter in response to Ricin^{CS} overexpression in the patched domain (Response Fig) As expected we did not see any changes to the GFP levels, which strongly suggest that reduction of miR-184 in response to Ricin^{CS} is responsible for the increase of dilp8-3'UTR reporter levels.*

Comment 3. Table S1. This table is confusing. In the Table S1 legend it says that the mean pupariation time is plotted, but no mean is shown. Rather a median is presented. Please clarify. If it is the median, please clarify what the median pupariation time represents. Is it the median of all larvae of pulled experiments or the median of medians or average of medians from N experiments? This is important as these values are not necessarily the same. Based on which values was the SEM calculated? SEM is related to an average value, not the median. Can you please clarify? It would be much preferable to include another dispersion value of the median pupariation time, if these are for pulled larvae from different experiments (e.g., 25-75th percentiles). Or show the average of the medians \pm SEM in addition to the median. As it is, the table is not completely useful. No need to show so many decimal cases. It would be fine to round it to 1 decimal case.

Response: *We thank the reviewers for pointing out this error. What we meant as median in the previous version of the table is the time taken for 50% of larvae to pupate. We have made the necessary changes, now it is clearly mentioned that the mean value of 50% pupariation time and SEM are shown in the respective columns.*

Comment 4. Figure Legend 1: there is conflicting info in A and B, in A it says that there are four independent repeats. After it says that "Data from five independent experiments are shown for A and B; ($p < 0.0001$)."

Please clarify.

Response: *We thank the reviewers for pointing out this error and have made the necessary changes.*

Comment 5. Figure 2A and B. Please check the error bars. They are not symmetric, which shouldn't happen in a linear Y axis.

Response: *We apologise for the error, and are thankful to the reviewers for pointing this out. We have made necessary corrections in figures 2A and 2B.*

Comment 6. Figure 2A - phm. Please verify the values and calculations. It is impossible that the average of the displayed three dots correspond to an average of 1, so something is wrong either with the normalization, the data, the bar, or the scale? I suspect the same problem occurs in the all other values - eg, for dib, there is no way the average is 1 there (values are $>0.5 + >0.5 + >2.0$, which will surely give >1 as average). Please revise the calculations and normalizations used for plotting the graphs (and verify if they were used for statistics). This seems like a systemic problem present in all panels reporting results from qPCR experiments.

Response: *We apologise for the error, and are thankful to the reviewers for pointing this out. We have made necessary corrections in all the figures as recommended.*

Comment 7. Figs. 4A,B; 5A,C,D; EV1; EV3; EV5B; EV6A,B,D & E. Again please verify error bars (they are not symmetrical) and the normalization/averaging of the data. Something is wrong

either with the data, the bar, or the scale. For reference, please compare to Fig. 2C (Ecdysone measurements in larvae), which shows reasonable dispersion of the values around the mean value of the bar, with accordingly symmetrical error bars.

Response: We apologise for the error, and are thankful to the reviewers for pointing this out. Updated figures are uploaded, as suggested.

Comment 8. Please add the controls (GAL4 and UAS alone) presented in the point-by-point response to the respective graphs in the manuscript, with the adjusted corresponding statistical analyses.

Response: We thank the reviewers for their suggestion and have revised the figures to include the same.

Minor comments

1. Results, Page 8, 2nd paragraph "The imaginal discs have been shown to regulate pupariation timing, especially in response to growth perturbations.." Please include seminal references that show this. eg, work by Pat Simpson (Simpson et al 1980), and Alex Shingleton's lab (eg, Stieper et al., 2008).

Response: We thank the reviewers for pointing this out and have now correctly revised the text to include the same.

2. Methods:

Pupariation assays: please indicate the number of repeats here as well apart from the legends. Please indicate the real range of animals used. It says 50, but Fig1A for instance says 142 and 104 animals for two genotypes with 4 independent repeats, so they cannot have been repeats of 50 animals. Same for Fig 1B, which indicates 955 and 617 animals for "five independent experiments", which means they had much more than 50 animals each.

Response: We would like to point out that 50 larvae we collected in each vial containing standard food to ensure optimum growth. As we were collecting larvae for the pupariation experiment within one hour of hatching we were not able to eliminate the larvae with balancers like TM6B and cyoGFP immediately. We therefore marked the pupariation timepoints of the larvae of the appropriate genotype with the help of balancers like TM6B or cyoGFP wheresoever homozygous flies were not present for crosses. We collected multiple vials of larvae over multiple days with 50 larvae in each vial and scored for the appropriate genotype.

3. Methods,

Fly stocks: something is grammatically missing in the last sentence of this paragraph.

Response: We thank you for pointing this out and apologise for this error. We have now revised the text

4. Methods,

Different symbols for degrees are used. Please standardize.

Response: We thank the reviewers for pointing this out. We have now revised the symbols as recommended.

5. Methods,

Sample collection for RNA isolation. Here it says that larvae were collected 116 hrs post-hatching, but in the Figure 2 it says 112 hrs. Please revise.

Response: We thank the reviewers for pointing this out and apologise for missing this. We have now correctly revised the text.

6. Methods,

qRT-PCR: here it says that "The level of the genes was normalized to the level of actin." However, in the Figure 2 (and other qRT-PCR figures), for instance, it says that they were normalized to rp49. No actin primers are described. Please revise. Please explain data analyses and plotting more clearly here.

Response: *We thank the reviewers for pointing this out and apologise for missing this. We have now corrected the text.*

7. Methods, microRNA qRT-PCR: please briefly explain the procedure and conditions used, including normalization. (eg, 2SrRNA is not mentioned here). The methods should stand alone and this is important for reproducibility, as kit instructions might be or become unavailable for others.

Response: *We thank the reviewers for pointing this out and will add in more details to ensure reproducibility.*

Thank you for the opportunity to revise your work. With kind regards,
Alisson Gontijo and Rebeca Zanini

Response: *Thank you for your comments and suggestions which helped us to improve our manuscript significantly.*

Reviewer 3: The authors have thoroughly addressed all my comments. In particular, the new results concerning the 3'UTR sensor (and its mutated version) significantly improve the manuscript. I therefore recommend its publication in Development.

Response: *We thank the reviewer for the kind recommendation.*

Second decision letter

MS ID#: dev.205280R1

MS Title: miR-184 modulates dilp8 to control developmental timing during normal growth conditions and in response to developmental perturbations

Authors: Jervis Fernandes, Muhammed Naseem, Ayisha Marwa and Jishy Varghese
 Article Type: Research Article

Dear Dr Varghese,

I am happy to tell you that your manuscript has been accepted for publication in Development, pending our standard publication integrity checks.

Reviewer 1

I recommend the article publication in Development.

Reviewer 2

The authors have addressed most of our comments, but have neither added new data nor text changes to address the limitations regarding the claim of a direct interaction between miR-184 and the dilp8-3'UTR, despite the fact that the authors acknowledge in their response to our comments that they cannot rule out insertional artefacts. We think therefore that the data presented do not fully justify the conclusion of direct regulation. We strongly recommend addressing this with a statement of limitation of the study acknowledging alternative explanation(s) before publication. Apart from this, we spotted a few minor points which could be easily rectified in a final text.

Major point:

Page 20, 2nd Paragraph, last sentence. "...confirming that the dilp8 gene is a target for post-transcriptional regulation by miR-184." and Discussion, page 23, 2nd paragraph, sentences "Our experiments identified dilp8... as a direct target of miR-184." and "Furthermore, we show that miR-184 directly regulates dilp8 expression via its 3' untranslated region (UTR) post-transcriptionally." We do not think these conclusions of direct regulation are fully supported by the data presented, so we strongly suggest the authors down-tone these conclusions to "suggestions/indications" and add a cautionary note on alternative explanations to these conclusions, as suggested and justified in our previous comments. Specifically, with their data, the authors cannot rule out the possibility that the tub-GFP-dilp8-3'UTR construct responds to miR-184 ***indirectly***. It is critical here that the conclusions on a supposed direct regulation rely on a single insertion, not multiple ones, of a construct, which is inserted in a different place from the only control, the construct tub-GFP-dilp8-mut-3'UTR. We are therefore not fully convinced by the data or the justifications provided in the authors' response.

This is the sole datapoint that would indicate direct regulation. All other results presented in this study can be explained by, for instance, a pathway where miR-184 regulates dilp8 indirectly:

miR-184 -| things that up-regulate dilp8 -> Dilp8 -> pupariation delay

If the insertion site of the tub-GFP-dilp8-3'UTR (say, site "A") responds to or similarly to e.g., any of the many "things that up-regulate dilp8", and the other site where the mutated version of this construct is inserted (say, site "B") does not, then the model of direct interaction between miR-184 and dilp8, even though it would be the simplest explanation for the result the authors observe, would be wrong. Hence, we strongly believe it is in the interest of both the authors and the journal readership that this limitation be acknowledged.

Minor points:

Results

Page 17, "...pupariation, a critical stage when the larva enters the metamorphosis stage under the influence of ecdysone hormone)".

Formally, pupariation is not the timepoint when the larva enters metamorphosis -Â that would be pupation, which occurs ~12h later. Hence, the sentence is not entirely precise. Can the authors revise it? Pupariation refers to a transitional stage where the puparium is formed. Formally, the larvae has not undergone ecdysis yet, so it is still a third instar larvae (not a pupa yet). Also, there is a lingering (closing) parenthesis after "hormone".

Page 18, third paragraph, "was also reduced in miR-184-depleted larvae (Fig. 2B)". Maybe change to "were also reduced in..."?

Page 19, third paragraph "Recent studies report that adult imaginal discs in larvae play". Maybe delete "adult"? "Imaginal" already refers to adult (imago).

Table S1. In the fourth column, according to the figure legend, the authors calculated the SEM of 50% pupariation TIME (hph), so the SEM is "in time (hours)" OR "in 50% pupariation time (hph)", not "in % pupariation", correct? Also, please write the P value as described in the legend, not "0" and be consistent with the annotations, i.e., if all are <0.0001 , then why show the value for miR-184, which is also <0.0001 . Also in the figure legend it says all comparisons are <0.0001 , but miR-375 says 0.0001. Is this rounded? If it is <0.0001 , then put <0.0001 .

Fig S2. Legend, first line, there is an extra "and", instead of a comma?.

Reviewer 3

SUMMARY OF THE ADVANCE MADE IN THIS PAPER AND ITS POTENTIAL SIGNIFICANCE TO THE FIELD

The authors have answered the comments from the reviewers. I believe the article is now ready for publication.